# Language-Assisted Feature Transformation for Anomaly Detection

**EungGu Yun**[†]
SAIGE
Seoul, South Korea

**Heonjin Ha**[‡]
LG Uplus
Seoul, South Korea

**Yeongwoo Nam**[‡]
Alsemy Inc.
Seoul, South Korea

**Bryan Dongik Lee**[‡]
Independent
Seoul, South Korea

## Abstract

This paper introduces LAFT, a novel feature transformation method designed to incorporate user knowledge and preferences into anomaly detection using natural language. Accurately modeling the boundary of normality is crucial for distinguishing abnormal data, but this is often challenging due to limited data or the presence of nuisance attributes. While unsupervised methods that rely solely on data without user guidance are common, they may fail to detect anomalies of specific interest. To address this limitation, we propose **L**anguage-**A**ssisted **F**eature **T**ransformation (LAFT), which leverages the shared image-text embedding space of vision-language models to transform visual features according to user-defined requirements. Combined with anomaly detection methods, LAFT effectively aligns visual features with user preferences, allowing anomalies of interest to be detected. Extensive experiments on both toy and real-world datasets validate the effectiveness of our method.

## 1 Introduction

Anomaly detection (AD) is the task of distinguishing abnormal data that deviates from the norm. In most scenarios where anomaly detection is applied, normal data is relatively easy to obtain, while abnormal data is scarce or sometimes impossible to obtain in advance. Thus, typical anomaly detection methods rely on normal data provided by users to learn what constitutes normal. However, when the training data is biased or does not cover the diverse variations, modeling the boundary of normality becomes a significant challenge (Lee & Wang, 2020; Cohen et al., 2023). In practical applications, models may need to prioritize or disregard certain attributes of the data. For instance, when inspecting products in images, a user might focus solely on the product's shape, ignoring attributes like color or lighting conditions. Moreover, distinguishing anomalies becomes more difficult when attributes are entangled, as seen in the Waterbirds dataset, where the background and bird features are entangled (Sagawa et al., 2019).

To address this issue, various methods have been proposed that use data augmentation or generation techniques to improve the learning of decision boundaries (Zavrtanik et al., 2021; Li et al., 2021; Du et al., 2021). These approaches aim to produce more diverse samples, covering a broader range of the underlying data distribution than what is available. Additionally, some approaches focus on enabling models to learn task-specific feature representations (Chen et al., 2020a;b; Caron et al., 2020), applying them to anomaly detection to better capture feature-level normality (Hyun et al., 2023). However, a limitation of these methods is that they may struggle to generalize to completely unseen data or fail to align with the user's intent in defining normality.

In some scenarios, users may have prior knowledge or specific preferences about the data that they want to integrate into the anomaly detection process. Typically, this is achieved through indirect methods, such as manually applying random color augmentation to ignore certain object colors. Controlling the boundary of normality remains relatively unexplored, and existing approaches often require unrealistic conditions, such as access to anomaly samples or labels (Cohen et al., 2023). To overcome this limitation, we propose leveraging vision-language models to directly integrate user knowledge and preferences into the anomaly detection framework through natural language. By

---

[†]Corresponding author: eg.yun@saige.ai
[‡]This work was primarily conducted while the authors were at SAIGE.

**Figure 1:** High-level motivation of our method: (**left**) typical image anomaly detection methods treat all test data that differs from the training data as anomalies, while (**right**) our method, LAFT AD, incorporates user preferences into the anomaly detection.

using language, users can more explicitly express their desired concepts, providing greater control over the definition of normality.

Recent studies have shown the effectiveness of training vision models with large amounts of unlabeled Internet data (Radford et al., 2021; Jia et al., 2021; Desai et al., 2023). By using image-text pairs from the web for pre-training, these models use natural language descriptions to improve the quality of image representations. Through large-scale training, they can correlate visual concepts in images with textual descriptions, aligning image and text features in a shared embedding space. Researchers have applied these models to industrial anomaly detection (Jeong et al., 2023; Cao et al., 2023b; Chen et al., 2023; Zhu & Pang, 2024) and general image out-of-distribution tasks using zero-shot text prompts (Ming et al., 2022; Miyai et al., 2023). A key benefit is the ability to incorporate human knowledge through text prompts, allowing zero-shot use without requiring additional training images. However, defining the complex normality of an image solely through natural language remains a challenge, and many methods face structural limitations in utilizing available training images. Therefore, we aim to develop a method where normality is primarily defined by image features, similar to other image anomaly detection approaches, with language serving only to refine the boundaries of normality.

In this paper, we present Language-Assisted Feature Transformation (LAFT), a method that allows users to control the transformation of image features using natural language without requiring additional training. LAFT leverages the vision-language model CLIP (Radford et al., 2021), using its shared embedding space to link visual and textual features. This connection enables the transformation of visual features based solely on language inputs. We hypothesize that visual concept subspaces exist within this shared embedding space, and introduce the notion of a "concept axis" to represent these subspaces. By computing pairwise differences between textual features, we derive concept difference vectors that define the concept axes. Projecting visual features onto or orthogonal to these axes allows for selective emphasis or suppression of specific image attributes.

Our method offers a training-free approach by using language, whereas most feature transformation methods require extensive training on data. This property presents two key advantages: it is agnostic to downstream tasks and performs well even in settings where data is scarce. By combining LAFT with proper anomaly detection methods, we can apply LAFT to anomaly detection tasks. Our approach differs from most work using CLIP for anomaly detection in that it relies primarily on image features to define normality, with language playing a supporting role. By using language, users can provide their understanding of normality, allowing greater flexibility in incorporating domain knowledge. Furthermore, by defining the boundaries of normality using image features, the model can accurately distinguish between normal and abnormal images.

We summarize our contributions as follows:

1. We propose Language-Assisted Feature Transformation (LAFT), a novel method that uses natural language to transform image features to fit the given task requirements by leveraging the image-text aligned embedding space of CLIP.
2. We introduce LAFT AD, an anomaly detection method that combines LAFT with a $k$-nearest neighbor ($k$NN) classifier, enabling users to selectively focus on or ignore specific image attributes based on their guidance for semantic anomaly detection tasks.
3. We present WinCLIP+LAFT, an extension of WinCLIP that integrates LAFT to improve performance in industrial anomaly detection tasks.
4. We demonstrate the effectiveness of our method on Colored MNIST and extensively evaluate its performance on real-world datasets, including Waterbirds, CelebA, MVTec AD, and VisA.

## 2   RELATED WORK

**Image anomaly detection with vision-language model**   Since the advent of CLIP (Radford et al., 2021), numerous studies in image anomaly detection have attempted to exploit the generalization capability of this vision-language model. To align visual and textual features for effective out-of-distribution detection, Ming et al. (2022) proposed a scoring method called MCM, with Miyai et al. (2023) later presenting an improved version, GL-MCM. Addressing the limitations of zero-shot, Ming & Li (2023) tried to further enhance performance by using parameter-efficient fine-tuning in downstream tasks. Fort et al. (2021) aimed to improve the model's understanding of normality by providing the CLIP text encoder with candidate anomaly labels. In addition, Esmaeilpour et al. (2022) introduced a framework for training a label generator based on the CLIP image encoder to generate possible anomaly labels. For industrial anomaly detection, Jeong et al. (2023) introduced WinCLIP, a zero-/few-shot anomaly detection model that efficiently extracts and aggregates features at multiple levels, aligning them with textual information. Similarly, Chen et al. (2023) proposed APRIL-GAN, and Zhu & Pang (2024) developed InCTRL, both of which adapt CLIP image features using additional adapter layers to better align them for anomaly detection, although these approaches require additional pre-training of the adapter layers.

**Adjusting the normality boundary**   Few studies have specifically addressed the challenge of adjusting the normality boundary in anomaly detection. Cohen et al. (2023) introduced Red PANDA, an anomaly detection method that disentangles relevant attributes in images while ignoring nuisance factors. However, achieving this disentangled feature representation requires labeled data for each nuisance attribute. Reiss et al. (2023) emphasized that overly expressive feature representations can ultimately degrade performance, highlighting a trade-off between sufficient representation and over-expressiveness in anomaly detection. Hendrycks et al. (2018) introduced outlier exposure, an approach that uses auxiliary data to help models generalize more effectively to unseen anomalies.

**Extracting task-specific features**   Several strategies have been developed to enhance the adaptability and robustness of features extracted from backbone models. Some studies focus on fine-tuning pre-trained feature extraction backbones or generating task-specific features through feature transformations. Ruff et al. (2018) introduced a method that transforms normal data into a hypersphere representation for anomaly detection, and Reiss et al. (2021) proposed an early stopping strategy to prevent feature collapse. Chen et al. (2020a;b) utilized contrastive pre-training to facilitate feature agreement, and Caron et al. (2020) employed prototype vectors for contrastive training of similar features. Following this line of research, Hyun et al. (2023); Reiss & Hoshen (2023); Tack et al. (2020) extended contrastive learning approaches for anomaly detection. Zhao et al. (2023) suggested using the backbone of vision-language pre-trained diffusion models and training a text adaptor to extract task-specific features with text prompts for downstream tasks.

## 3   PRELIMINARIES

In our scenario, a training set, represented as $\mathcal{D}_{\text{train}}$, consists of normal samples only, and a test set $\mathcal{D}_{\text{test}}$ consists of both normal and anomalous samples. For a two-stage anomaly detection model consisting of a feature extractor $f$ and an anomaly classifier $g$, the feature extractor $f$ maps the input image $x$ to a feature $v = f(x)$, and the anomaly classifier $g$ maps the feature $v$ to an anomaly score $s = g(v)$. Then, the anomaly score $s_i$ is used to determine the prediction of the anomaly label $\hat{y}_i$.

The attributes of an image $x$ extracted by the feature extractor are denoted as $a = \{a^1, \cdots, a^m\}$, and the anomaly label is denoted as $y$. Each attribute $a^j$ $(j = 1, \cdots, m)$ denotes any characteristics within the feature extracted from the image, such as the shape of the object, the color, or the background. The $m$ attributes can be divided into relevant attributes $a^{\text{rel}} = \{a^j\}_{1 \leq j \leq n}$ and irrelevant (nuisance) attributes $a^{\text{irr}} = \{a^j\}_{n < j \leq m}$ for desired anomaly detection tasks. For example, when detecting anomalies in the shape of objects, the shape is relevant, while the color is irrelevant. To properly detect anomalies, the prediction of the model should be invariant to the irrelevant attributes.

There are two ways to achieve this invariance:

1. Provide enough data that covers the possible values for each $a^j$, so that the classifier $g$ can properly ignore irrelevant attributes $a^{\text{irr}}$ in the feature $v$. This is the most desirable solution, and many data augmentation and generation methods have been proposed. However, it is often impossible to collect or hard to generate such data.

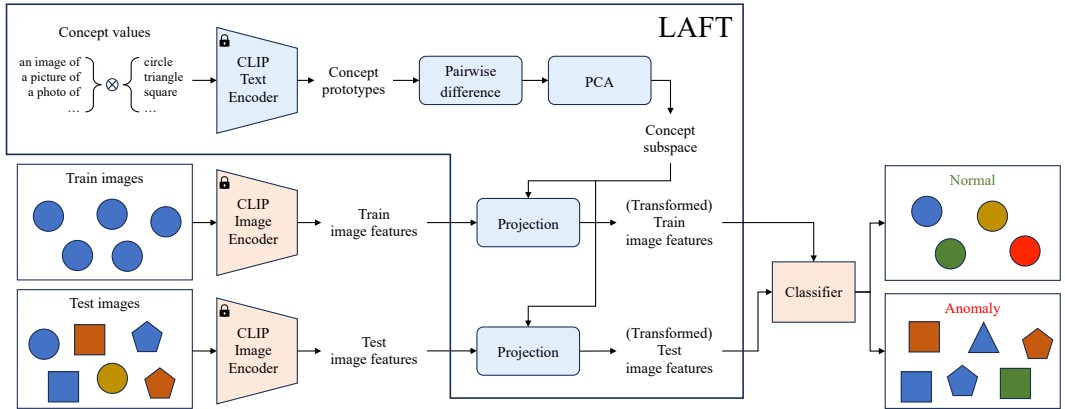

**Figure 2:** Overview of our method, LAFT, a transformation module, and LAFT AD, combining LAFT with a $k$NN classifier. Our approach uses CLIP's text and image encoders without any additional training. The key idea is to use text prompts containing concept values to construct a concept subspace for the target attribute. This process involves computing pairwise differences of concept prototypes and extracting robust concept axes via PCA. Once the concept subspaces are created, the shared embedding space can be used to transform image features suitable for anomaly detection.

2. Make the feature extractor $f$ only extract the relevant attributes $a^{\text{rel}}$ and do not include the irrelevant attributes $a^{\text{irr}}$ in the feature $v$. Fine-tuning the feature extractor or adding a transformation $T$ to the feature extractor is a common way to achieve this. But it is often hard to design such training procedures to achieve the invariance.

Our goal is to design a transformation $T$ that transforms the feature $v$ into a new feature $v' = T(v)$ that the anomaly classifier $g$ can use to detect anomalies without being affected by the irrelevant attributes $a^{\text{irr}}$ using only the user's natural language without any additional training data or labels.

This can be achieved by two approaches:

**Guide** Make a transformation $T_{\text{guide}}$ that includes only the relevant attributes $a^j \in a^{\text{rel}}$. Some attributes in $a^{\text{rel}}$ may be correlated, so the transformed feature may not include all relevant attributes.

**Ignore** Make a transformation $T_{\text{ignore}}$ that excludes all irrelevant attributes $\forall a^j \in a^{\text{irr}}$. In many cases this is harder to achieve than the above approach, because the transformation should be able to remove all irrelevant attributes.

That is, we want our transformation $T$ to represent the relevant attributes in a manner unaffected by the irrelevant attributes:

$$p(a^{n+1}, \cdots, a^m) = p(a^{n+1}, \cdots, a^m \,|\, T(v)). \tag{1}$$

We also want the transformed feature $v'$ to be informative, containing enough information about relevant attributes. Here, $I(;)$ represents the mutual information between the two arguments:

$$I((a^1, \cdots, a^n); v) \sim I((a^1, \cdots, a^n); T(v)). \tag{2}$$

In practice, invariance can be measured by the accuracy of predicting the anomaly label $\hat{y}$ from the transformed feature $T(v)$. But we can assess the informativeness by measuring the accuracy of predicting the relevant attribute utilized to define anomalies. Empirical evaluations of these measures for various datasets are provided in the Experiments. With such a representation, anomalies can later be evaluated independently, devoid of any bias caused by the irrelevant attribute we aim to disregard.

CLIP (Radford et al., 2021) embeds the features in a unit sphere subspace in Euclidean space $\mathbb{R}^n$. An embedding vector of an image is correlated to the text embedding describing the image. This means that we can construct the transform with the CLIP text encoder. We assume that all relevant and irrelevant features can be encoded with the text description, so that natural language assists in the manipulation of the vector in the CLIP shared embedding space.

# 4 METHOD

Anomaly detection often faces data scarcity, especially for abnormal data, making it difficult for models to define the normality boundary. In this situation, users may have prior knowledge or preferences about what should be considered normal. Therefore, we aim to design:

- A method that can be used when the user has knowledge of normality and wants to control it. Typical anomaly detection methods consider all test data different from the training data as anomalies, but we want our method to be used only with a language.

- A method that effectively handles anomalies that are challenging to express solely in natural language. Other methods using vision-language models require normality to be expressed entirely in language prompts for guidance, which limits the ability to capture complex normality. We want our method to utilize image features to define normality.

Note that our method is not intended to be used in situations where the user has no knowledge or preferences, or to automatically identify relevant attributes.

Our method uses CLIP's text and image encoders without additional training. The core idea is to construct a **concept subspace** for target attributes using text prompts containing **concept values**. After constructing the subspace, it transforms image features by projecting them onto the subspace, taking advantage of CLIP's shared embedding space. This involves computing pairwise differences between textual features containing **concept prototypes** to extract robust concept axes. In this section, we provide a detailed explanation of our method, as illustrated in Figure 2.

## 4.1 TEXT PROMPT

To guide the model in focusing on or ignoring specific attributes of an image, it is essential to provide it with appropriate textual prompts. Following Ming et al. (2022), we assume that the text contains **concept prototypes** representing the attributes. Thus, we provide the method with a list of prompts composed of templates and values, as commonly done in CLIP-based methods (Radford et al., 2021; Ming et al., 2022; Jeong et al., 2023). The key difference in our approach is that we use the actual values of the desired attribute (e.g., "circle," "square") rather than the attribute name itself (e.g., "shape"). For instance, to capture the concept of hair color, we can construct the prompt as:

- "a photo of a person with *brown hair*"
- "a potrait of a man with *black hair*"
- "an image of a *blond* child"

By using the actual values of the desired attribute in the prompts, we aim for the method to capture the difference between the concept prototypes of the attribute. Providing values for this attribute that are not present in the training set, but are likely to appear during testing, helps construct a more comprehensive subspace for that concept. As with other language-based methods, multiple types of templates can be provided to mitigate the bias introduced by any single template. We examine the effect of various sets of concept values in the Ablation Study, with the prompts used in our experiments detailed in the Appendix.

## 4.2 FIND CONCEPT SUBSPACE

Mikolov (2013) showed that simple arithmetic operations between text embeddings can capture meaningful relationships (e.g., vec(biggest) − vec(big) ≈ vec(smallest) − vec(small)). This finding showed that text embeddings not only represent texts in a vector space, but also encode the underlying relationships between them. Moreover, they observed that high-dimensional vectors, when trained on large datasets, are capable of capturing subtle semantic relationships. Similarly, CLIP's text embeddings support arithmetic operations to compute differences between concept prototypes, allowing for the comparison of these concepts in the embedding space.

Building on this approach, the method constructs a subspace of the concept within CLIP's embedding space. Specifically, it identifies the axes of this subspace that capture the variance between concept prototypes, as represented by difference between the prompts. For prompts $t_i$ and $t_j$, where $1 \leq i < j \leq n$, we compute the pairwise differences of the text features:

$$\Delta v_{ij} := E_{\text{text}}(t_i) - E_{\text{text}}(t_j) \tag{3}$$

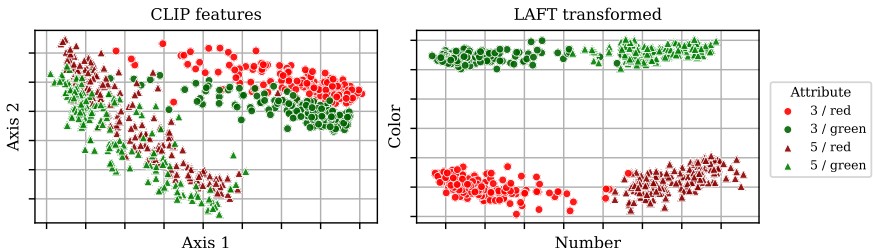

**Figure 3:** Projection of image features from CLIP's image encoder (**left**) and transformed image features using LAFT (**right**). Without guidance, the image features may not align with the intended attributes. After applying LAFT, the features become more aligned with the desired attributes.

where $n$ represents the number of prompts, and $E_{\text{text}}$ denotes CLIP's text encoder. But directly using these vectors is not preferable because the prompts may contain template noise (Zhou et al., 2022). To address this, we apply PCA to extract the principal axes from these vectors:

$$\{c_k\}_{1 \le k \le d} := \text{PCA}(\{\Delta v_{ij}\}_{1 \le i < j \le n}, d) \tag{4}$$

where $d$ is the number of components, and $\{c_k\}$ represents the $d$ principal axes, collectively referred to as the ***concept axes***. Throughout this paper, we typically select $d$ between 8 and 32 when guiding an attribute, and between 32 and 384 when ignoring an attribute. As discussed in the Preliminaries, ignoring attributes is generally more challenging than guiding them, so a larger number of components is used. For the impact of $d$, please refer to the Ablation Study.

### 4.3 FEATURE TRANSFORMATION WITH PROJECTION

For each image feature $v_i = f(x_i)$ encoded by CLIP's image encoder, we project the features onto the concept subspace:

$$v_i' = T_{\text{guide}}(v_i) := \sum_{k=1}^{d} \frac{\langle v_i, c_k \rangle}{\langle c_k, c_k \rangle} c_k, \tag{5}$$

where $\langle \cdot, \cdot \rangle$ denotes the inner product. This projection retains only the relevant attributes of the image feature, as irrelevant attributes are nearly orthogonal to the concept axes. Conversely, we can remove the irrelevant attributes using orthogonal projection. Let $\bar{c}_k$ represent the concept axes associated with the irrelevant attributes. Then we can project orthogonally onto the concept subspace as follows:

$$\bar{v}_i' = T_{\text{ignore}}(v_i) := v_i - \sum_{k=1}^{d} \frac{\langle v_i, \bar{c}_k \rangle}{\langle \bar{c}_k, \bar{c}_k \rangle} \bar{c}_k, \tag{6}$$

which manually cancels out the vectors of irrelevant attributes. This completes the description of the feature transformation method, LAFT.

### 4.4 ANOMALY SCORING

Many anomaly detection methods employ $k$-nearest-neighbors ($k$NN) for anomaly scoring (Cohen & Hoshen, 2020; Roth et al., 2022). This approach is effective because normal data tends to be densely concentrated, while anomalous data is typically sparsely distributed in the feature space. In our method, LAFT AD, we also use $k$NN to estimate the density of normal data around each test sample, assuming that the features have been processed by LAFT for *semantic anomaly detection*.

We start by extracting features for each normal sample: $v_i' = T(f(x_i)), \forall x_i \in \mathcal{D}_{\text{train}}$. Next, for each test sample, we infer its feature: $v_j' = T(f(x_j)), \forall x_j \in \mathcal{D}_{\text{test}}$. Finally, we score each test sample based on its $k$NN distance from the normal data:

$$s_j = g(v_j') := \frac{1}{k} \sum_{v_j' \in N_k(v_i')} S_{\cos}(v_j', v_i'), \tag{7}$$

where $N_k(v_i')$ denotes the $k$ nearest features to $v_j'$ in the normal data, and $S_{\cos}(\cdot, \cdot)$ represents cosine similarity. We use $k = 30$ for $k$NN throughout the paper without optimization.

However, this method may not be suitable for industrial anomaly detection tasks, where anomalies are often small and subtle. To address this, we extend WinCLIP (Jeong et al., 2023) to incorporate LAFT for anomaly scoring, which we refer to as WinCLIP+LAFT for *industrial anomaly detection*. A detailed explanation of this extension is provided in the Experiment.

## 5 EXPERIMENTS

**Datasets**  To validate our approach, we used the colored version of MNIST (LeCun et al., 2010), Waterbirds (Sagawa et al., 2019), and CelebA (Liu et al., 2015) datasets for semantic anomaly detection (SAD). We defined normal and anomalous values for each dataset attribute and divided the training split into $2^m$ subsets. For example, in the Colored MNIST dataset, we designated digits 0-4 as normal and 5-9 as anomalous, with the color red as normal and green and blue as anomalous. We then used one subset as the training set, considering it normal across all $m$ attributes (e.g., digits 0-4 and the color red). This is similar to the setup commonly used in many studies for a single attribute (primarily based on class labels) (Ruff et al., 2020; Tack et al., 2020; Esmaeilpour et al., 2022; Cohen et al., 2023; Reiss & Hoshen, 2023; Cao et al., 2023a; Zhu & Pang, 2024). To further demonstrate the practicality of our method, we also used the MVTec AD (Bergmann et al., 2019) and VisA (Zou et al., 2022) datasets for industrial anomaly detection (IAD).

**Baselines**  For semantic anomaly datasets, we do not compare with typical image-only AD methods for two reasons: (1) they require attribute-specific processing (e.g., color augmentation) to incorporate user prior knowledge, which limits generalizability to other contexts, and (2) without guidance, these methods detect images that differ from the training set, resulting in high false positive rates in our settings. Instead, to simulate image-only AD methods, we compare with simple **kNN** and **LinearProbe** with additional training data, directly using image features from CLIP.

**kNN** computes the distance between the test image features and the training image features for anomaly scoring. As discussed in the Method, many AD methods rely on **kNN**-based scoring, making it an important baseline. **kNN** using the same training subset as the other methods serves as a no-guidance version of LAFT AD, allowing us to directly evaluate LAFT's effectiveness. And **kNN** using additional normal training subset depending on the target attribute represents an image-only method with attribute-specific image processing. Since applying image augmentation across all datasets and attributes is not straightforward (e.g., augmenting the background in Waterbirds), we assume that the additional data is well *augmented* images for the target attribute. To evaluate CLIP image encoder's performance, we provide full training data including normal and anomalous images for LinearProbe to train a linear classifier to predict the class of the test image (Radford et al., 2021).

We also evaluate CLIP-based zero-shot and few-shot AD methods. For zero-shot AD, we use Maximum Concept Matching (**MCM**; Ming et al., 2022), which requires only prompts for normal images to perform anomaly scoring, and Zero-shot outlier exposure (**ZOE**; Fort et al., 2021), which uses prompts for normal images and candidate prompts for anomalous images. And **CLIPN** (Wang et al., 2023) is a zero-shot method that uses pre-trained "no" prompts and "no" text encoder to make text features for anomalies. We also consider **WinCLIP** (Jeong et al., 2023), which supports both zero-/few-shot AD. The zero-shot version of WinCLIP is similar to ZOE in terms of anomaly scoring, and the few-shot version (WinCLIP+) requires a few normal images. Lastly, we consider **InCTRL** (Zhu & Pang, 2024), a few-shot AD method similar to WinCLIP+.

**Prompts**  We use the actual class names from the dataset for concept values, if available, and add other candidate labels to simulate unseen classes. For example, for the number attribute in the Colored MNIST dataset, we use '0' to '20' and 'zero' to 'twenty' as number attributes, even though the dataset only includes 0 to 9. We referenced the prompts provided by CLIP (Radford et al., 2021).

For more details, please refer to the Experimental Details.

### 5.1 SEMANTIC ANOMALY DETECTION

We used the colored version of the MNIST dataset (LeCun et al., 2010), similar to Arjovsky et al. (2019), to demonstrate our concept in the simplest way. We created a dataset that divides each digit of the MNIST and colors each split with red, green, and blue. In this way, the image of a colored MNIST consists of two attributes: number and color. We mark the numbers 0 to 4 as normal and the numbers 5 to 9 as abnormal. In addition, we label red as normal and green and blue as abnormal colors. In this setting, the training set consists of 0 to 4 and red images. Then, we use five different seeds to split the training set for coloring each digit. Figure 3 shows a brief overview of our desired transformation using concept axes. If we choose an axis (number or color) to project the image features, we can simply use **kNN** to detect only the desired anomalies.

**Table 1:** Anomaly detection performance (%) on Colored MNIST and Waterbirds datasets. Standard deviations are computed over five different seeds, with results for deterministic cases omitted. The best values are shown in **bold**, and the second-best values are underlined.

| | | Colored MNIST: Number | | | Waterbirds: Bird | | |
|---|---|---|---|---|---|---|---|
| Guidance | Method | AUROC ↑ | AUPRC ↑ | FPR95 ↓ | AUROC ↑ | AUPRC ↑ | FPR95 ↓ |
| **Baseline** | | | | | | | |
| Subset of normal images | $k$NN | $92.4 \pm 0.2$ | $91.8 \pm 0.2$ | $31.9 \pm 0.6$ | 82.3 | 91.2 | 48.1 |
| + All normal images | $k$NN | $98.0 \pm 0.0$ | $97.1 \pm 0.0$ | $7.5 \pm 0.2$ | 83.0 | 91.5 | 44.7 |
| + Anomalous images | LinearProbe | $99.8 \pm 0.0$ | $99.8 \pm 0.0$ | $0.5 \pm 0.1$ | $91.0 \pm 0.0$ | $96.7 \pm 0.0$ | $34.2 \pm 0.0$ |
| **Guide** | | | | | | | |
| Language | MCM | 62.9 | 52.5 | 60.8 | 88.8 | 95.4 | 40.0 |
| | ZOE | 91.2 | 92.4 | 47.3 | 92.2 | 97.1 | 32.8 |
| | CLIPN-C | $73.0 \pm 2.5$ | $61.7 \pm 3.0$ | $51.0 \pm 0.9$ | $71.2 \pm 2.8$ | $86.5 \pm 1.1$ | $100.0 \pm 0.0$ |
| | CLIPN-A | $73.2 \pm 2.2$ | $61.6 \pm 2.7$ | $50.0 \pm 0.6$ | $82.3 \pm 0.8$ | $91.9 \pm 0.3$ | $55.6 \pm 1.2$ |
| | WinCLIP | 91.1 | 92.4 | 48.0 | 92.2 | 97.0 | 32.6 |
| Image + Language | WinCLIP+ | $92.6 \pm 1.3$ | $91.3 \pm 2.0$ | $38.8 \pm 1.5$ | $91.8 \pm 0.2$ | $96.9 \pm 0.1$ | $33.4 \pm 1.5$ |
| | InCTRL | $94.0 \pm 1.3$ | $92.4 \pm 2.6$ | $25.5 \pm 4.1$ | $83.6 \pm 1.0$ | $92.0 \pm 0.7$ | $63.5 \pm 3.1$ |
| | LAFT AD (Ours) | $\mathbf{98.5} \pm 0.0$ | $\mathbf{98.4} \pm 0.0$ | $\mathbf{6.9} \pm 0.1$ | **95.6** | **98.4** | **20.6** |
| **Ignore** | | | | | | | |
| Image + Language | LAFT AD (Ours) | $\underline{97.4} \pm 0.1$ | $\underline{96.9} \pm 0.2$ | $\underline{10.4} \pm 0.4$ | 84.8 | 92.2 | 38.6 |

**Table 2:** Anomaly detection performance (%) on CelebA dataset. Standard deviations are computed over five different seeds, with results for deterministic cases omitted. The best values are shown in **bold**, and the second-best values are underlined.

| | | Hair color | | | Eyeglasses | | |
|---|---|---|---|---|---|---|---|
| Guidance | Method | AUROC ↑ | AUPRC ↑ | FPR95 ↓ | AUROC ↑ | AUPRC ↑ | FPR95 ↓ |
| **Baseline** | | | | | | | |
| Subset of normal images | $k$NN | 83.3 | 96.6 | 62.4 | 83.0 | 21.6 | 47.7 |
| + All normal images | $k$NN | 83.4 | 96.6 | 62.4 | 85.3 | 22.0 | 43.2 |
| + Anomalous images | LinearProbe | $98.2 \pm 0.0$ | $99.7 \pm 0.0$ | $9.6 \pm 0.0$ | $99.7 \pm 0.0$ | $98.4 \pm 0.0$ | $0.1 \pm 0.0$ |
| **Guide** | | | | | | | |
| Language | MCM | 84.5 | 97.2 | 68.4 | 5.7 | 3.3 | 100.0 |
| | ZOE | 93.9 | 99.0 | 35.7 | 82.6 | 31.5 | 67.3 |
| | CLIPN-C | $82.8 \pm 1.7$ | $96.8 \pm 0.4$ | $72.0 \pm 2.4$ | $1.4 \pm 0.1$ | $3.7 \pm 0.0$ | $100.0 \pm 0.0$ |
| | CLIPN-A | $84.7 \pm 1.2$ | $97.2 \pm 0.3$ | $70.2 \pm 2.1$ | $1.2 \pm 0.1$ | $3.4 \pm 0.0$ | $100.0 \pm 0.0$ |
| | WinCLIP | 93.7 | 98.9 | 37.7 | 83.6 | 34.6 | 66.2 |
| Image + Language | WinCLIP+ | $92.8 \pm 0.3$ | $98.8 \pm 0.1$ | $41.2 \pm 1.4$ | $85.0 \pm 2.4$ | $26.8 \pm 3.0$ | $47.8 \pm 6.9$ |
| | InCTRL | $85.7 \pm 0.9$ | $96.9 \pm 0.3$ | $67.8 \pm 1.6$ | $\underline{87.8} \pm 1.6$ | $30.4 \pm 2.9$ | $\underline{29.6} \pm 4.4$ |
| | LAFT AD (Ours) | **95.0** | **99.2** | **29.8** | **98.1** | **80.7** | **5.9** |

Table 1 presents the main results on Colored MNIST dataset with the target attribute being the number. The table is divided into three groups: *baseline*, *guide* (Lang., Img. + Lang.), and *ignore*, as discussed in the Preliminaries. The *baseline* group serves as a reference point for performance that relies solely on images, as mentioned earlier. Its performance varies depending on the amount of image data available to the model. The *guide* group consists of methods that can be instructed to focus on a target attribute, where models are given prompts related to the attribute corresponding to the label (e.g., number prompts for number anomalies). Specifically, methods in the *guide* (Lang.) group rely solely on language, while those in the *guide* (Img. + Lang.) group use both image and language to define normality. However, except for our method, language guidance in these methods is used only to calculate image-text similarity and is not applied to image-image similarity. The *ignore* group represents a method that disregards attributes other than the target, where models are provided prompts of irrelevant attributes (e.g., color prompts for number anomalies). Ignoring irrelevant attributes is a unique feature of our method, but it is generally a more challenging task.

As shown in the table, guided methods (*guide* (Img. + Lang.)) generally outperform non-guided baselines ($k$NN with a subset of normal images), with our method achieving the best overall performance. The performance of guided methods that use only language (*guide* (Lang.)) is lower than that of the baseline methods because they are provided with inaccurate prompts for anomalous images (e.g., '13'). This problem, highlighted in Ming et al. (2022), shows that methods relying on image-text

**Table 3:** Anomaly detection AUROC (%) on MVTec AD and VisA datasets in few-shot settings. We use five different sets of reference samples from the training set. $K$ denotes the number of reference samples. The best values are shown in **bold**, and the second-best values are underlined.

| Method | MVTec AD | | | | | VisA | | | | |
|---|---|---|---|---|---|---|---|---|---|---|
| | $K=0$ | $K=1$ | $K=2$ | $K=4$ | $K=8$ | $K=0$ | $K=1$ | $K=2$ | $K=4$ | $K=8$ |
| InCTRL | — | $91.3 \pm 2.7$ | $93.2 \pm 1.8$ | $93.6 \pm 1.6$ | $93.8 \pm 1.3$ | — | $\underline{85.0} \pm 4.3$ | $\underline{86.7} \pm 2.7$ | $\mathbf{88.4} \pm 2.0$ | $\mathbf{89.4} \pm 1.7$ |
| WinCLIP/+ | 90.4 | $93.5 \pm 1.6$ | $95.2 \pm 0.7$ | $95.6 \pm 0.5$ | $95.7 \pm 0.9$ | 75.5 | $83.4 \pm 2.8$ | $85.6 \pm 1.8$ | $86.8 \pm 1.9$ | $88.2 \pm 1.2$ |
| + LAFT-G (Ours) | **92.6** | $\underline{94.7} \pm 1.4$ | $\mathbf{96.1} \pm 0.5$ | $\underline{96.2} \pm 0.3$ | $\underline{96.4} \pm 0.4$ | $\underline{80.0}$ | $85.0 \pm 1.3$ | $86.1 \pm 1.3$ | $87.0 \pm 1.5$ | $88.2 \pm 1.1$ |
| + LAFT-C (Ours) | $\underline{92.5}$ | $\mathbf{94.8} \pm 1.4$ | $\underline{96.0} \pm 0.5$ | $\mathbf{96.3} \pm 0.4$ | $\mathbf{96.5} \pm 0.5$ | **80.6** | $\mathbf{85.7} \pm 1.4$ | $\mathbf{87.0} \pm 1.1$ | $\underline{87.4} \pm 1.4$ | $\underline{88.3} \pm 1.0$ |

similarity in CLIP are highly sensitive to inaccurate prompts. WinCLIP performs similarly to ZOE because multi-scale features do not benefit semantic anomaly detection. While WinCLIP+ performs better than WinCLIP by using reference images, its performance is still below ours.

The Waterbirds dataset is widely used in studies of spurious correlations and disentangling representations. It consists of two primary attributes: bird type (waterbird / landbird) and background (water / land). Naturally, the training set has a very strong correlation between birds and backgrounds, whereas the test set has an equal ratio of birds to backgrounds. We specify waterbirds and water backgrounds as the normal training set. Table 1 summarizes the results, where the target attribute is the bird type. The trends observed in the Colored MNIST experiment are largely consistent, demonstrating the applicability of our method to real-world datasets. The key difference is that ignoring one attribute (background) does not directly improve performance on the other attribute (bird).

To verify that our method works in multi-attribute datasets, we use the CelebA dataset, which contains over 200K celebrity images with 40 attribute labels. For the normal training set, we select two attributes: Hair color and Eyeglasses. The results are displayed in Table 2, where the target attributes are Hair color and Eyeglasses. The trends are consistent with the previous experiments, demonstrating the effectiveness of our method.

The performance of CLIPN and InCTRL was inconsistent across different datasets and target attributes, suggesting that their generalization ability is lower than that of CLIP, likely due to the presence of modules trained on specific datasets. In contrast, our method uses pre-trained CLIP without any additional training, making it more generalizable. Additionally, our method effectively guides one attribute while ignoring the others, as discussed further in the Additional Experiments.

## 5.2 INDUSTRIAL ANOMALY DETECTION

To demonstrate the practical applicability of our method beyond semantic anomaly detection, we evaluated its performance on the widely used MVTec AD (Bergmann et al., 2019) and VisA (Zou et al., 2022) datasets in few-shot settings. However, anomalies in industrial anomaly detection datasets often consist of small defects that are difficult to distinguish using only image-level representations. Instead of using LAFT AD, which is designed for semantic anomaly detection tasks, we propose WinCLIP+LAFT, a model that applies LAFT to WinCLIP to extract multi-scale features using CLIP. We apply LAFT to WinCLIP's window, image, and text embeddings, all of which reside in CLIP's shared embedding space, allowing seamless integration.

Typically, some zero-shot or few-shot methods based on CLIP rely on training additional adapter layers to transform CLIP's image features for anomaly detection tasks. For example, InCTRL pre-trains feature adapters on specific datasets (such as MVTec AD) to effectively compute the similarity before applying them to different datasets. In contrast, our method uses prompts to transform image features while preserving CLIP's core features, allowing us to extract features suitable for anomaly detection tasks without the need for additional training of the adapter layer.

For the proof of concept, we used prompts similar to those in WinCLIP for **LAFT General** (LAFT-G) and more category-specific prompts for **LAFT Category** (LAFT-C). For LAFT General, we constructed prompts using only state words and category names, without providing additional knowledge based on the category of the inspection image as in WinCLIP. However, in order to identify a more precise concept subspace, we used more text templates and general state words such as 'malformed {}'. For LAFT Category, we used prompts that include category-specific knowledge (e.g. anomaly class names), such as 'bottle with large breakage' for the bottle category.

**Table 4:** Anomaly detection performance (%) on the Colored MNIST and Waterbirds datasets with various prompts. Standard deviations are computed over five different seeds. The best values are shown in **bold**, and the second-best values are underlined.

| Prompt | Concept values | | | Colored MNIST: Number | | | Waterbirds: Bird | | |
| | Seen | Unseen | Aux. | AUROC ↑ | AUPRC ↑ | FPR95 ↓ | AUROC ↑ | AUPRC ↑ | FPR95 ↓ |
|---|---|---|---|---|---|---|---|---|---|
| **Guide** | | | | | | | | | |
| Only normals | ○ | × | × | $96.2 \pm 0.2$ | $95.7 \pm 0.2$ | $16.6 \pm 0.2$ | 94.3 | 97.8 | 22.6 |
| Partial anomalies | ○ | △ | × | $98.4 \pm 0.0$ | $98.3 \pm 0.0$ | $8.2 \pm 0.1$ | 95.5 | 98.3 | **18.7** |
| Exact anomalies | ○ | ○ | × | $\textbf{98.8} \pm 0.0$ | $\textbf{98.8} \pm 0.1$ | $\textbf{6.2} \pm 0.1$ | **95.9** | **98.6** | 19.6 |
| All candidates | ○ | ○ | ○ | $\underline{98.5} \pm 0.0$ | $\underline{98.4} \pm 0.0$ | $6.9 \pm 0.1$ | 95.6 | 98.4 | 20.6 |
| **Ignore** | | | | | | | | | |
| Only seen normals | ○ | × | × | $94.5 \pm 0.2$ | $94.1 \pm 0.4$ | $25.0 \pm 0.5$ | 84.8 | **92.4** | 40.6 |
| Partial unseen normals | ○ | △ | × | $95.9 \pm 0.2$ | $95.1 \pm 0.3$ | $16.7 \pm 0.4$ | **85.1** | **92.4** | 38.1 |
| Exact normals | ○ | ○ | × | $\underline{97.2} \pm 0.1$ | $96.5 \pm 0.3$ | $\underline{11.6} \pm 0.3$ | 85.0 | 92.2 | **37.3** |
| All candidates | ○ | ○ | ○ | $\textbf{97.4} \pm 0.1$ | $\textbf{96.9} \pm 0.2$ | $\textbf{10.4} \pm 0.4$ | 84.8 | 92.2 | 38.6 |

The results are presented in Table 3, which summarizes the average performance across all categories. As shown in the results, WinCLIP+LAFT consistently outperforms WinCLIP in both zero-shot and few-shot scenarios. Furthermore, our method achieves superior or comparable performance to InCTRL, a pre-trained model for industrial anomaly detection, without requiring additional training. See the Full Results for detailed results on each category.

### 5.3 ABLATION STUDY ON PROMPT QUALITY

An important consideration when using LAFT is how to provide the user's prior knowledge. In anomaly detection, we generally have a good understanding of the current training data, but the unseen test data remains unknown. Therefore, we investigated how the performance of LAFT changes depending on the quality of the concept values provided, as shown in Table 4. In the table, *Seen* refers to the concept values for the current training data, *Unseen* refers to the concept values for the unseen test data, and *Aux.* denotes concept values that are not present in the dataset.

For example, in Colored MNIST, if the guiding attribute is the number, *Seen* represents the values 0-4, *Unseen* corresponds to 5-9, and *Aux.* refers to values like 10-20. Similarly, if the ignored attribute is color, *Seen* includes red, *Unseen* covers green and blue, and *Aux.* includes colors such as yellow and purple. The symbol ○ indicates that all concept values are used, while △ indicates that only half of the concept values are utilized.

The results show that the performance of LAFT is robust to the quality of the concept values when at least partial concept values are provided. Furthermore, the performance is not significantly affected when the completely not included concept values are provided. Also, providing concept values that are not included at all does not significantly affect the performance. These characteristics show that LAFT can be effectively used in anomaly detection where only limited information is known.

## 6 CONCLUSION

In this paper, we introduce Language-Assisted Feature Transformation (LAFT), a novel feature transformation method designed to integrate user knowledge and preferences into the anomaly detection framework via natural language, without the need for additional data or training. By utilizing the shared embedding space of the vision-language model, LAFT can align visual features with user-provided texts to guide or ignore specific attributes in the image. We also presented LAFT AD, an anomaly detection method that integrates LAFT with a $k$NN classifier, and WinCLIP+LAFT, an extension of WinCLIP that incorporates LAFT for industrial anomaly detection. This combination allows users to adjust the normality boundary of the model by providing texts to detect desired anomalies. Through experiments on synthetic and real-world datasets, we demonstrate the effectiveness of our proposed method.

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

## A    LIMITATIONS AND DISCUSSION

**Table 5:** Anomaly detection performance (%) on the CelebA dataset. We transform the image and text features using LAFT to ignore the Gender attribute. Then, we use the transformed features for the MCM and ZOE methods. Underlined numbers indicate the performance of the suppressed attribute, and **bold** numbers indicate the performance of the non-suppressed attribute.

| Method | Hair color | | | Gender | | |
|--------|----------|----------|-----------|----------|----------|-----------|
|        | AUROC ↑ | AUPRC ↑ | FPR95 ↓ | AUROC ↑ | AUPRC ↑ | FPR95 ↓ |
| MCM    | 88.0 | 97.9 | 56.7 | 96.9 | 98.0 | 13.7 |
| + LAFT | **93.8** | **99.0** | **35.4** | 32.4 | 50.0 | 98.4 |
| ZOE    | 93.4 | 98.9 | 38.8 | 99.4 | 99.6 | 02.5 |
| + LAFT | **95.1** | **99.2** | **30.6** | 50.0 | 61.2 | 94.7 |

**Ignore attributes using LAFT**    Unlike in a simple Colored MNIST dataset, we observe that ignoring one attribute using LAFT does not directly improve the performance of the other attribute in real-world datasets. However, as seen in Appendix C, the LAFT actually suppresses the attribute to be ignored. As mentioned in the Preliminaries, it is hard to remove all attribute-related information in the embedding space using only text prompts. Alternatively, guiding the attribute is relatively easy, because LAFT only needs to capture the primary information about the attribute.

**Selecting the number of PCA component $d$**    LAFT introduces a hyperparameter $d$, which represents the number of PCA components. While the performance remains relatively stable within a certain range, as demonstrated in Appendix C, selecting an appropriate value remains crucial. However, determining the optimal $d$ is particularly challenging in the context of unsupervised anomaly detection, where ground truth labels are unavailable. Consequently, $d$ must be selected heuristically. Developing an automatic selection mechanism for $d$ would be a valuable direction for future research to improve the usability of LAFT.

**Using LAFT with other methods**    LAFT can be used not only for anomaly detection, but also as a feature transformation module in other tasks or methods. Basically, we expect that it can be applied to any vision model that requires a feature extractor. For a simple example, we apply the LAFT method to MCM and ZOE. The results in Table 5 show that we can suppress the Gender attribute. Applying LAFT to other downstream tasks would be a future work.

## B    EXPERIMENTAL DETAILS

We provide the source code of our method at `https://github.com/yuneg11/LAFT`.

**Backbones**    For semantic anomaly detection datasets (Colored MNIST, Waterbirds, CelebA), we use the CLIP ViT-B/16 (Radford et al., 2021) with the pre-trained checkpoint from Fang et al. (2023). And since InCTRL (Zhu & Pang, 2024) and CLIPN (Wang et al., 2023) require pretrained weights, we used the network with the pretrained weights provided by the authors. For industrial anomaly detection datasets (MVTec AD), we use the CLIP ViT-B/16+ (Gadre et al., 2024), pre-trained on the LAION-400M (Schuhmann et al., 2021) dataset, following the setup used in WinCLIP (Jeong et al., 2023). For a fair comparison, we also adopted the CLIP's image encoder as a feature extractor for the $k$NN baseline.

**Metrics**    We use three metrics to evaluate the performance of the methods: the Area Under Receiver Operating Characteristics (AUROC), the Area Under the Precision Recall Curve (AUPRC), and the False Positive Rate at the 95% true positive rate (FPR95). AUROC and FPR95 are commonly used for the anomaly detection or out-of-distribution detection task (Ming et al., 2022). And we also use AUPRC because some datasets are imbalanced, with a significant disparity.

**Computing resources**    We use a single NVIDIA RTX 3090 GPU for all experiments.

**Hyperparameter**    The only hyperparameter in LAFT is the number of PCA components $d$. We typically choose $d$ from $\mathbf{4}$ to $\mathbf{32}$ when guiding an attribute and from $\mathbf{32}$ to $\mathbf{384}$ when ignoring an attribute. Refer to the Ablation Study for the impact of $d$ on the performance. And we use $k = \mathbf{30}$ for the methods using $k$NN anomaly scoring ($k$NN and LAFT AD).

**Prompts**    To see the actual prompts used in the experiments, please refer to the `laft/prompts` in the source code. We referenced the prompts provided by CLIP (Radford et al., 2021) [1].

- **Colored MNIST**
  - **Number**: "zero", ..., "twenty", "0", ..., "20"
  - **Color**: "red", "green", "blue", "yellow", "orange", ..., "black", "white"

- **Waterbirds**
  - **Bird**: Class names provided by the dataset and Birdsnap class names.
  - **Color**: "land", "bamboo", "forest", "ocean", and similar words.

- **CelebA**
  - **Hair color**: "blond", "black", "brown", "gray", "red", "white", and similar words.
  - **Eyeglasses**: "glasses", "eyeglasses", "sunglasses"
  - **Gender**: "man", "male", "boy", "woman", "female", "girl", "masculine", "feminine"

- **MVTec AD** and **VisA**    We use the same prompts as WinCLIP (Jeong et al., 2023) for anomaly scoring. To compute the LAFT concept subspace, we use some more template-/state-level prompts for both LAFT General and LAFT Category. For LAFT Category, we use additional category-level prompts as Li et al. (2024).
  - **Template-level**: Jeong et al. (2023) and "an image of a {}", "a photo of the {}", ...
  - **State-level**: Jeong et al. (2023) and "{} in perfect condition", "malformed {}", ...
  - **Category-level (LAFT Category)**: "bottle with large breakage", "carpet with hole", ...

**Dataset Split**

- **Colored MNIST**    `R` denotes red, `G` denotes green, and `B` denotes blue colored digits. `0-4` and `5-9` denote the digits from 0 to 4 and from 5 to 9, respectively.
  - **Train**: `R/0-4` (16.67%)
  - **Test**: `R/0-4` (16.67%), `R/5-9` (16.67%), `GB/0-4` (33.33%), `GB/5-9` (33.33%)

- **Waterbirds**    `Wbird` denotes waterbirds, and `Lbird` denotes landbirds. `Wback` denotes water background, and `Lback` denotes land background.
  - **Train**: `Wbird/Wback` (22.04%)
  - **Test**: `Wbird/Wback` (11.08%), `Wbird/Lback` (11.08%), `Lbird/Wback` (38.92%), `Lbird/Lback` (38.92%)

- **CelebA**    `Blond` denotes blond hair, and `Glass` denotes eyeglasses. `-Blond` denotes non-blond hair, and `-Glass` denotes no eyeglasses.
  - **Train**: `Blond/-Glass` (14.66%)
  - **Test**: `Blond/Glass` (13.01%), `Blond/-Glass` (0.31%), `-Blond/Glass` (80.53%), `-Blond/-Glass` (6.15%)

- **MVTec AD** and **VisA**    We use the same split as Bergmann et al. (2019) and Zou et al. (2022).

---

[1] `https://github.com/openai/CLIP/blob/main/data/prompts.md`

## C  ADDITIONAL EXPERIMENTS

### C.1  GUIDING AND IGNORING ATTRIBUTES

**Table 6:** Anomaly detection performance (%) on the Colored MNIST datasets with different target criteria. We use five different seeds to split the training set for coloring each digit. **Bold** numbers indicate that the performance should be high (relevant), and underlined numbers indicate that the performance should be low (irrelevant).

| Criteria | Number | | | Color | | |
|---|---|---|---|---|---|---|
| | AUROC ↑ | AUPRC ↑ | FPR95 ↓ | AUROC ↑ | AUPRC ↑ | FPR95 ↓ |
| **No guidance** | | | | | | |
| $k$NN | $89.7 \pm 0.4$ | $89.5 \pm 0.4$ | $44.0 \pm 0.3$ | $81.1 \pm 0.6$ | $89.5 \pm 0.5$ | $58.1 \pm 0.2$ |
| **Guide** | | | | | | |
| Number | $\mathbf{98.5} \pm 0.0$ | $\mathbf{98.4} \pm 0.0$ | $\mathbf{06.9} \pm 0.1$ | $\underline{52.7} \pm 0.1$ | $\underline{66.4} \pm 0.2$ | $\underline{89.3} \pm 0.2$ |
| Color | $\underline{51.2} \pm 0.1$ | $\underline{53.6} \pm 0.1$ | $\underline{94.8} \pm 0.1$ | $\mathbf{100.0} \pm 0.0$ | $\mathbf{100.0} \pm 0.0$ | $\mathbf{0.0} \pm 0.0$ |
| **Ignore** | | | | | | |
| Number | $\underline{63.7} \pm 0.2$ | $\underline{65.6} \pm 0.2$ | $\underline{75.8} \pm 0.1$ | $\mathbf{99.9} \pm 0.0$ | $\mathbf{99.9} \pm 0.0$ | $\mathbf{0.2} \pm 0.0$ |
| Color | $\mathbf{97.4} \pm 0.1$ | $\mathbf{96.9} \pm 0.2$ | $\mathbf{10.4} \pm 0.4$ | $\underline{60.2} \pm 0.4$ | $\underline{73.0} \pm 0.5$ | $\underline{78.0} \pm 0.1$ |

**Table 7:** Anomaly detection performance (%) on the Waterbirds dataset. Standard deviations are not reported because the method is deterministic. **Bold** numbers indicate that the performance should be high (relevant), and underlined numbers indicate that the performance should be low (irrelevant).

| Criteria | Bird | | | Background | | |
|---|---|---|---|---|---|---|
| | AUROC ↑ | AUPRC ↑ | FPR95 ↓ | AUROC ↑ | AUPRC ↑ | FPR95 ↓ |
| **No guidance** | | | | | | |
| $k$NN | 82.3 | 91.2 | 48.1 | 68.0 | 63.9 | 87.3 |
| **Guide** | | | | | | |
| Bird | **95.3** | **98.2** | **19.5** | 70.6 | 72.6 | 87.2 |
| Background | 59.5 | 84.5 | 91.4 | **97.6** | **97.2** | **10.1** |
| **Ignore** | | | | | | |
| Bird | 63.9 | 59.3 | 87.2 | **74.5** | **87.6** | **62.6** |
| Background | **84.8** | **92.2** | **38.6** | 52.7 | 50.0 | 92.6 |

The results in Table 6 and Table 7 show the performance of two attributes when one attribute is either guided or ignored. In datasets with two clearly distinguishable attributes, such as Colored MNIST, ignoring one attribute implicitly guides the other. However, in datasets composed of real-world images, such as Waterbirds, although there may appear to be only two attributes, there may actually be many more. As a result, while the performance of non-ignored attributes may improve slightly, the overall improvement is not significant. Nevertheless, when an attribute is ignored, we can confirm that it is indeed properly disregarded.

### C.2  ABLATION STUDY ON PCA COMPONENTS

To investigate the impact of the number of PCA components $d$ on the performance of LAFT, we conduct an ablation study on semantic anomaly datasets. The results are shown in Table 8 and Table 9. We observe that the performance of LAFT is not very sensitive to the number of PCA components for a certain range of $d$. However, the performance drops significantly when $d$ is too small or too large. This is because a small $d$ cannot capture the concept subspace well, while a large $d$ may include irrelevant information. Except for Eyeglasses attribute in CelebA, the best performance is achieved when $d$ is between 14 and 28 for guiding attributes. For the Eyeglasses attribute, the best performance is when $d$ is 6, which we expect because it is a simple binary classification problem of whether a person wears glasses or not, rather than distinguishing between different types within an attribute (like distinguishing between different numbers or different types of birds).

**Table 8:** Anomaly detection performance (%) on Colored MNIST and Waterbirds datasets. We scan the number of PCA components $d$, which is the only hyperparameter of LAFT module. The best values are shown in **bold**, and the second-best values are underlined.

| $d$ | Colored MNIST: Number | | | Waterbirds: Bird | | |
|---|---|---|---|---|---|---|
| | AUROC ↑ | AUPRC ↑ | FPR95 ↓ | AUROC ↑ | AUPRC ↑ | FPR95 ↓ |
| **Guide** | | | | | | |
| 2 | 80.3 | 79.5 | 73.7 | 87.2 | 95.0 | 67.7 |
| 4 | 93.5 | 94.2 | 42.3 | 91.2 | 95.9 | 35.7 |
| 6 | 96.3 | 96.3 | 19.3 | 93.6 | 97.8 | 26.4 |
| 8 | 97.2 | 97.3 | 14.7 | 93.9 | 97.8 | 25.5 |
| 10 | 97.4 | 97.3 | 13.2 | 94.2 | 97.9 | 23.8 |
| 12 | 97.7 | 97.7 | 11.1 | 94.6 | 98.1 | 22.4 |
| 14 | 97.8 | 97.7 | 10.0 | 95.1 | 98.2 | 21.8 |
| 16 | 98.0 | 97.7 | 10.5 | 95.4 | 98.2 | 21.0 |
| 18 | 98.2 | 97.8 | 9.0 | **95.6** | **98.4** | 20.6 |
| 20 | 98.4 | 97.9 | 8.8 | **95.6** | 98.3 | 20.3 |
| 24 | **98.5** | **98.2** | 8.4 | 95.5 | 98.2 | 20.1 |
| 28 | 98.4 | 98.1 | **7.9** | 95.4 | 98.2 | **19.3** |
| 32 | 98.3 | 98.0 | 8.6 | 95.0 | 98.0 | 21.3 |
| 40 | 97.8 | 97.6 | 10.5 | 94.9 | 97.9 | 21.1 |
| 48 | 97.7 | 97.5 | 10.7 | 94.4 | 97.6 | 21.4 |
| 64 | 97.1 | 96.8 | 13.1 | 93.5 | 97.1 | 23.0 |
| **Ignore** | | | | | | |
| 8 | 96.2 | 95.3 | 16.2 | 82.6 | 91.1 | 41.7 |
| 16 | 96.8 | 95.9 | 13.3 | 82.6 | 91.1 | 42.0 |
| 32 | 97.1 | 96.2 | 12.0 | 83.4 | 91.4 | 40.8 |
| 64 | 97.1 | 96.4 | 11.9 | 83.4 | 91.4 | 40.1 |
| 96 | 97.2 | 96.4 | 11.8 | 83.5 | 91.4 | 40.2 |
| 128 | 97.3 | 96.6 | 11.0 | 83.4 | 91.4 | 40.1 |
| 160 | **97.4** | 96.7 | 11.0 | 83.4 | 91.4 | 40.0 |
| 192 | **97.4** | 96.8 | 11.0 | **84.8** | **92.2** | **38.6** |
| 224 | **97.4** | **96.9** | **10.6** | 83.8 | 91.7 | 41.8 |
| 256 | 97.2 | 96.7 | 11.9 | 83.8 | 91.7 | 42.4 |
| 288 | 97.1 | 96.6 | 12.5 | 82.3 | 91.0 | 44.0 |
| 320 | 96.9 | 96.3 | 13.6 | 81.6 | 90.6 | 45.6 |
| 352 | 96.7 | 96.1 | 15.1 | 81.6 | 90.7 | 44.6 |
| 384 | 96.4 | 96.1 | 17.0 | 81.5 | 91.0 | 46.4 |

**Table 9:** Anomaly detection performance (%) on CelebA dataset. We scan the number of PCA components $d$, which is the only hyperparameter of LAFT module. The best values are shown in **bold**, and the second-best values are underlined.

| $d$ | Hair color | | | Eyeglasses | | |
|---|---|---|---|---|---|---|
| | AUROC ↑ | AUPRC ↑ | FPR95 ↓ | AUROC ↑ | AUPRC ↑ | FPR95 ↓ |
| **Guide** | | | | | | |
| 2 | 52.6 | 88.4 | 95.3 | 65.9 | 11.4 | 86.3 |
| 4 | 90.5 | 98.3 | 55.8 | 97.3 | 76.4 | 8.2 |
| 6 | 92.6 | 98.7 | 38.4 | **98.1** | **80.7** | **5.9** |
| 8 | 94.0 | 99.0 | 36.5 | 97.9 | 78.6 | 6.8 |
| 10 | 94.6 | 99.1 | 31.9 | 97.5 | 75.0 | 8.3 |
| 12 | 94.8 | 99.1 | 31.7 | 97.6 | 77.0 | 9.4 |
| 14 | **95.0** | **99.2** | 29.8 | 97.6 | 78.3 | 9.8 |
| 16 | 94.8 | 99.1 | 30.6 | 97.3 | 75.3 | 11.1 |
| 18 | 94.7 | 99.1 | **29.5** | 96.8 | 72.0 | 13.5 |
| 20 | 94.6 | 99.1 | 30.5 | 95.3 | 60.1 | 18.1 |
| 24 | 94.3 | 99.1 | 31.7 | 95.2 | 57.1 | 18.6 |
| 28 | 94.0 | 99.0 | 33.9 | 94.8 | 55.7 | 21.0 |
| 32 | 93.9 | 99.0 | 34.4 | 94.2 | 47.4 | 20.6 |
| 40 | 93.3 | 98.9 | 38.0 | 92.5 | 40.6 | 24.5 |
| 48 | 92.8 | 98.8 | 38.5 | 91.4 | 38.4 | 28.5 |
| 64 | 91.8 | 98.6 | 40.9 | 89.7 | 32.7 | 31.8 |

## C.3 LAFT with Various Vison-Language Models

**Table 10:** Anomaly detection AUROC (%) on semantic anomaly datasets. We compare vision-language models across diverse backbone architectures and training strategies. The best values are shown in **bold**, and the second-best values are underlined.

| Method | CLIP ViT | | | CLIP ConvNeXt | | EVA02 | | SigLIP ViT | | CoCa ViT | |
|---|---|---|---|---|---|---|---|---|---|---|---|
| | B-16 | L-14 | H-14 | Base | XXL | B-16 | L-14 | B-16 | L-16 | B-32 | L-14 |
| **Colored MNIST: Number** | | | | | | | | | | | |
| *k*NN | 92.4 | 90.7 | 89.8 | 84.2 | 87.5 | 66.7 | 73.1 | 86.1 | 82.3 | 85.4 | 89.4 |
| MCM | 62.9 | 79.8 | 87.1 | 73.9 | 83.9 | 40.2 | 64.0 | 74.7 | 68.2 | 62.1 | 70.4 |
| ZOE | 91.2 | 90.7 | 93.3 | 92.7 | 96.2 | 83.4 | 92.7 | 93.0 | 95.9 | 93.6 | 97.4 |
| LAFT AD (Ours) | **98.5** | **98.9** | **99.6** | **97.6** | **99.5** | **89.0** | **94.3** | **98.5** | **99.1** | **98.7** | **98.8** |
| **Waterbirds: Bird** | | | | | | | | | | | |
| *k*NN | 82.3 | 87.1 | 85.9 | 79.7 | 85.9 | 86.0 | 88.7 | 83.6 | 79.2 | 76.0 | 84.8 |
| MCM | 88.8 | 94.3 | 94.7 | 85.2 | 92.6 | 88.8 | 94.9 | 89.7 | 93.8 | 81.5 | 87.4 |
| ZOE | 92.2 | 93.8 | 93.7 | 90.6 | 93.5 | 93.6 | 94.4 | 93.3 | 94.2 | 88.4 | 92.5 |
| LAFT AD (Ours) | **95.6** | **97.2** | **97.3** | **94.4** | **97.2** | **96.6** | **98.6** | **96.2** | **97.6** | **91.1** | **95.4** |
| **CelebA: Hair color** | | | | | | | | | | | |
| *k*NN | 83.3 | 85.3 | 83.7 | 89.7 | 88.0 | 88.3 | 86.0 | 85.4 | 83.4 | 90.4 | 89.2 |
| MCM | 84.5 | 87.9 | 87.2 | 87.9 | 87.4 | 86.7 | 88.9 | 86.8 | 82.1 | 85.1 | 85.0 |
| ZOE | 93.9 | 94.3 | 93.6 | 92.5 | 93.2 | 93.0 | 93.5 | 91.5 | 87.7 | 94.1 | 90.1 |
| LAFT AD (Ours) | **95.0** | **94.8** | **95.1** | **94.4** | **95.1** | **94.9** | **94.5** | **94.8** | **94.8** | **94.8** | **93.7** |
| **CelebA: Eyeglasses** | | | | | | | | | | | |
| *k*NN | 83.0 | 81.0 | 75.0 | 77.5 | 78.1 | 76.2 | 75.5 | 75.8 | 70.0 | 79.6 | 78.1 |
| MCM | 5.7 | 11.7 | 13.2 | 12.6 | 11.5 | 21.6 | 24.8 | 12.2 | 14.9 | 10.6 | 13.6 |
| ZOE | 82.6 | **98.8** | **98.9** | 94.8 | 93.9 | **98.7** | **97.9** | **99.2** | **99.2** | 84.9 | 91.1 |
| LAFT AD (Ours) | **98.1** | 97.9 | 98.1 | **97.4** | **98.1** | 97.9 | 97.6 | 98.1 | 98.1 | **98.1** | **97.6** |

We compare the performance of LAFT with vision-language models across diverse backbone architectures and training strategies on the semantic anomaly datasets. We use architectures with pre-trained weights available in OpenCLIP (Ilharco et al., 2021). Specifically, we use ViT (Dosovitskiy et al., 2021), ConvNeXt (Liu et al., 2022), and EVA02 (Fang et al., 2024) as the backbone architectures, and we use the pre-trained weights from CLIP (Radford et al., 2021), EVA-CLIP (Sun et al., 2023), SigLIP (Zhai et al., 2023), and CoCa (Yu et al., 2022). The results are shown in Table 10. We observed that across various vision-language models, LAFT AD consistently outperformed the baseline method in most cases.

## D Full Results on MVTec AD and VisA

We provide the complete results on the MVTec AD and VisA datasets in Table 11 and Table 12. As shown in these tables, LAFT significantly enhances the performance of both anomaly detection and localization in the zero-shot setting. Additionally, LAFT improves anomaly detection performance in the few-shot setting.

Despite these improvements, there are certain limitations to applying LAFT to industrial anomaly detection datasets. First, as discussed in the Limitations, the selection of $d$ relies on a heuristic approach, requiring manual tuning for each category. Second, LAFT does not improve anomaly localization performance. We hypothesize that this is due to the localization task requiring more fine-grained information than anomaly detection, while LAFT may remove subtle details that are not well captured by text prompts. Nonetheless, even in such cases, our method achieves anomaly localization performance that remains comparable to the original WinCLIP.

Developing more sophisticated approaches for applying LAFT to industrial anomaly detection remains an important direction for future research.

**Table 11:** Full anomaly detection and localization AUROC (%) on the MVTec AD dataset in few-shot settings (k-shot). We use five different sets of reference samples from the training set for each method. We **bold** the best performance and underline the second-best performance in each category.

**Anomaly Detection**

| # K | Method | Bottle | Cable | Capsule | Carpet | Grid | Hazelnut | Leather | Metal Nut | Pill | Screw | Tile | Toothbrush | Transistor | Wood | Zipper | Average |
|---|---|---|---|---|---|---|---|---|---|---|---|---|---|---|---|---|---|
| 0 | WinCLIP | 98.6 | 85.1 | 68.7 | 99.3 | 99.2 | 92.4 | 100. | 96.2 | 81.6 | 71.6 | 99.9 | 85.3 | 89.1 | 97.6 | 91.2 | 90.4 |
| | + LAFF-G (Ours) | 98.7 | 89.6 | 80.2 | 99.8 | 99.3 | 92.4 | 100. | 97.1 | 86.3 | 78.6 | 99.9 | 87.2 | 91.0 | 97.6 | 91.8 | 92.6 |
| | + LAFF-C (Ours) | 98.7 | 86.4 | 81.3 | 99.8 | 99.4 | 92.4 | 100. | 97.1 | 86.4 | 77.1 | 99.8 | 87.5 | 90.9 | 98.0 | 92.5 | 92.5 |
| 1 | InCTRL | 98.8 ± 0.9 | 83.3 ± 2.1 | 68.8 ± 9.9 | 99.4 ± 0.6 | 98.8 ± 1.3 | 97.4 ± 0.6 | 99.9 ± 0.1 | 95.3 ± 4.5 | 89.8 ± 2.4 | 74.3 ± 3.0 | 99.9 ± 0.1 | 96.2 ± 4.0 | 81.7 ± 1.3 | 97.0 ± 0.6 | 88.8 ± 8.7 | 91.3 ± 2.7 |
| | WinCLIP | 99.4 ± 0.1 | 89.7 ± 0.7 | 72.7 ± 8.9 | 99.6 ± 0.1 | 99.4 ± 0.4 | 96.8 ± 0.9 | 100. ± 0.0 | 98.6 ± 0.5 | 93.3 ± 0.8 | 77.9 ± 3.1 | 100. ± 0.0 | 95.0 ± 4.3 | 88.0 ± 0.6 | 99.3 ± 0.3 | 92.6 ± 3.9 | 93.5 ± 1.6 |
| | + LAFF-G (Ours) | 99.4 ± 0.2 | 91.4 ± 0.4 | 77.3 ± 9.8 | 99.8 ± 0.1 | 99.5 ± 0.2 | 96.8 ± 0.8 | 100. ± 0.0 | 98.6 ± 0.3 | 92.8 ± 0.7 | 82.2 ± 1.5 | 99.9 ± 0.0 | 96.3 ± 2.9 | 91.2 ± 0.2 | 99.3 ± 0.3 | 94.7 ± 3.8 | 94.7 ± 1.4 |
| | + LAFF-C (Ours) | 99.6 ± 0.1 | 89.7 ± 0.6 | 78.3 ± 9.9 | 99.8 ± 0.1 | 99.6 ± 0.3 | 96.9 ± 0.8 | 100. ± 0.0 | 98.7 ± 0.4 | 92.5 ± 0.6 | 83.1 ± 1.2 | 99.9 ± 0.0 | 96.2 ± 3.4 | 91.3 ± 0.1 | 99.3 ± 0.2 | 94.9 ± 2.8 | 94.8 ± 1.4 |
| 2 | InCTRL | 98.4 ± 0.9 | 87.0 ± 3.7 | 79.2 ± 5.2 | 99.6 ± 0.4 | 99.0 ± 1.0 | 97.8 ± 0.9 | 99.9 ± 0.1 | 95.5 ± 3.9 | 90.0 ± 1.4 | 77.0 ± 3.7 | 99.9 ± 0.0 | 97.8 ± 1.4 | 85.4 ± 3.4 | 97.1 ± 0.4 | 93.9 ± 0.8 | 93.2 ± 1.8 |
| | WinCLIP | 99.6 ± 0.1 | 90.3 ± 1.0 | 85.9 ± 2.3 | 99.6 ± 0.1 | 99.5 ± 0.4 | 97.8 ± 0.4 | 100. ± 0.0 | 99.0 ± 0.2 | 93.6 ± 0.5 | 81.1 ± 1.1 | 100. ± 0.0 | 96.9 ± 1.1 | 89.4 ± 1.5 | 99.3 ± 0.2 | 95.7 ± 0.9 | 95.2 ± 0.7 |
| | + LAFF-G (Ours) | 99.5 ± 0.2 | 91.8 ± 0.5 | 91.9 ± 2.0 | 99.8 ± 0.1 | 99.5 ± 0.3 | 97.6 ± 0.4 | 100. ± 0.0 | 99.1 ± 0.1 | 93.2 ± 0.4 | 82.7 ± 0.9 | 99.9 ± 0.1 | 97.7 ± 1.1 | 91.7 ± 0.2 | 99.3 ± 0.1 | 96.9 ± 0.5 | 96.1 ± 0.5 |
| | + LAFF-C (Ours) | 99.7 ± 0.1 | 89.9 ± 0.7 | 93.3 ± 1.9 | 99.8 ± 0.1 | 99.6 ± 0.3 | 97.7 ± 0.4 | 100. ± 0.0 | 99.0 ± 0.1 | 93.0 ± 0.5 | 81.0 ± 0.7 | 99.9 ± 0.0 | 97.7 ± 1.1 | 91.5 ± 0.6 | 99.3 ± 0.1 | 97.4 ± 0.3 | 96.0 ± 0.5 |
| 4 | InCTRL | 98.4 ± 1.0 | 88.9 ± 2.6 | 75.9 ± 5.7 | 99.6 ± 0.3 | 98.8 ± 1.0 | 97.7 ± 0.5 | 99.9 ± 0.1 | 97.1 ± 2.5 | 91.2 ± 1.4 | 78.3 ± 3.5 | 99.9 ± 0.0 | 98.7 ± 0.9 | 87.2 ± 2.7 | 97.4 ± 0.1 | 94.2 ± 0.9 | 93.6 ± 1.6 |
| | WinCLIP | 99.6 ± 0.3 | 90.7 ± 0.8 | 87.1 ± 0.7 | 99.6 ± 0.2 | 99.5 ± 0.3 | 98.2 ± 0.3 | 100. ± 0.0 | 99.2 ± 0.3 | 93.5 ± 0.2 | 82.4 ± 2.4 | 100. ± 0.0 | 98.2 ± 0.2 | 90.4 ± 1.2 | 99.4 ± 0.1 | 95.5 ± 1.1 | 95.6 ± 0.5 |
| | + LAFF-G (Ours) | 99.5 ± 0.1 | 90.1 ± 0.9 | 93.3 ± 0.9 | 99.9 ± 0.0 | 99.1 ± 0.5 | 97.9 ± 0.2 | 100. ± 0.0 | 99.2 ± 0.2 | 93.2 ± 0.2 | 83.0 ± 0.8 | 99.9 ± 0.0 | 98.4 ± 0.2 | 92.0 ± 0.2 | 99.3 ± 0.1 | 97.0 ± 0.5 | 96.2 ± 0.3 |
| | + LAFF-C (Ours) | 99.7 ± 0.0 | 90.5 ± 0.8 | 94.7 ± 0.7 | 99.8 ± 0.1 | 99.6 ± 0.3 | 98.0 ± 0.2 | 100. ± 0.0 | 99.2 ± 0.2 | 93.0 ± 0.3 | 81.7 ± 2.3 | 99.9 ± 0.0 | 98.4 ± 0.2 | 92.0 ± 0.4 | 99.4 ± 0.1 | 97.4 ± 0.4 | 96.3 ± 0.4 |
| 8 | InCTRL | 98.2 ± 0.7 | 91.2 ± 2.2 | 71.5 ± 4.7 | 99.8 ± 0.1 | 98.7 ± 0.8 | 97.5 ± 0.2 | 100. ± 0.0 | 97.3 ± 1.5 | 91.3 ± 1.2 | 81.2 ± 3.6 | 100. ± 0.0 | 98.7 ± 0.9 | 88.8 ± 1.8 | 97.6 ± 0.2 | 95.1 ± 0.7 | 93.8 ± 1.3 |
| | WinCLIP | 97.7 ± 2.3 | 92.1 ± 0.9 | 87.6 ± 0.9 | 99.8 ± 0.1 | 99.3 ± 0.2 | 98.1 ± 0.2 | 100. ± 0.0 | 99.3 ± 0.2 | 94.1 ± 0.3 | 85.6 ± 3.0 | 100. ± 0.0 | 94.9 ± 4.5 | 91.1 ± 0.6 | 99.4 ± 0.1 | 96.2 ± 0.7 | 95.7 ± 0.9 |
| | + LAFF-G (Ours) | 99.6 ± 0.1 | 91.4 ± 0.8 | 93.5 ± 0.7 | 99.9 ± 0.0 | 99.1 ± 0.3 | 97.8 ± 0.2 | 100. ± 0.0 | 99.3 ± 0.1 | 93.7 ± 0.2 | 84.6 ± 2.7 | 99.9 ± 0.1 | 98.5 ± 0.5 | 92.6 ± 0.2 | 99.3 ± 0.1 | 97.4 ± 0.4 | 96.4 ± 0.4 |
| | + LAFF-C (Ours) | 99.7 ± 0.1 | 91.5 ± 0.7 | 94.6 ± 0.7 | 99.9 ± 0.0 | 99.4 ± 0.1 | 97.9 ± 0.2 | 100. ± 0.0 | 99.3 ± 0.2 | 93.4 ± 0.3 | 84.8 ± 3.1 | 99.8 ± 0.1 | 98.4 ± 0.5 | 92.2 ± 0.3 | 99.4 ± 0.1 | 97.8 ± 0.5 | 96.5 ± 0.5 |

**Anomaly Localization**

| # K | Method | Bottle | Cable | Capsule | Carpet | Grid | Hazelnut | Leather | Metal Nut | Pill | Screw | Tile | Toothbrush | Transistor | Wood | Zipper | Average |
|---|---|---|---|---|---|---|---|---|---|---|---|---|---|---|---|---|---|
| 0 | WinCLIP | 85.7 | 61.3 | 87.0 | 90.9 | 79.4 | 95.7 | 95.5 | 49.3 | 72.7 | 91.1 | 79.1 | 86.2 | 83.7 | 85.1 | 91.7 | 82.3 |
| | + LAFF-G (Ours) | 90.0 | 68.3 | 84.9 | 94.7 | 82.0 | 95.6 | 96.3 | 51.7 | 87.1 | 81.9 | 87.5 | 89.6 | 86.7 | 89.9 | 92.0 | 85.2 |
| | + LAFF-C (Ours) | 90.1 | 66.7 | 91.2 | 94.4 | 82.3 | 95.9 | 96.3 | 52.2 | 85.2 | 88.4 | 85.9 | 88.1 | 83.5 | 88.3 | 94.2 | 85.5 |
| 1 | WinCLIP | 94.8 ± 0.3 | 89.8 ± 0.8 | 95.6 ± 0.8 | 99.0 ± 0.1 | 94.3 ± 1.0 | 98.5 ± 0.2 | 99.3 ± 0.0 | 78.1 ± 1.1 | 93.8 ± 0.2 | 96.1 ± 0.1 | 91.6 ± 0.5 | 96.2 ± 0.6 | 87.2 ± 1.0 | 94.6 ± 0.5 | 96.7 ± 0.2 | 93.7 ± 0.5 |
| | + LAFF-G (Ours) | 96.2 ± 0.1 | 81.0 ± 0.3 | 95.9 ± 0.6 | 99.0 ± 0.1 | 94.7 ± 0.7 | 98.3 ± 0.2 | 99.2 ± 0.0 | 73.3 ± 0.9 | 94.1 ± 0.2 | 94.4 ± 0.3 | 94.8 ± 0.3 | 95.4 ± 0.3 | 89.3 ± 0.3 | 95.3 ± 0.3 | 95.3 ± 0.5 | 93.1 ± 0.3 |
| | + LAFF-C (Ours) | 96.2 ± 0.1 | 86.0 ± 0.8 | 96.1 ± 0.6 | 98.9 ± 0.1 | 94.1 ± 0.8 | 98.3 ± 0.2 | 99.2 ± 0.0 | 74.4 ± 1.0 | 94.3 ± 0.2 | 80.6 ± 0.6 | 94.0 ± 0.3 | 95.2 ± 0.5 | 86.8 ± 0.6 | 95.4 ± 0.3 | 97.4 ± 0.1 | 92.5 ± 0.4 |
| 2 | WinCLIP | 95.0 ± 0.1 | 90.5 ± 0.7 | 96.9 ± 0.1 | 98.9 ± 0.1 | 94.9 ± 0.9 | 98.7 ± 0.1 | 99.3 ± 0.0 | 79.4 ± 1.3 | 94.1 ± 0.2 | 96.4 ± 0.2 | 91.8 ± 0.1 | 96.9 ± 0.5 | 89.3 ± 0.5 | 94.7 ± 0.4 | 97.0 ± 0.1 | 94.3 ± 0.4 |
| | + LAFF-G (Ours) | 96.4 ± 0.1 | 80.9 ± 0.3 | 97.0 ± 0.1 | 99.0 ± 0.1 | 95.0 ± 0.7 | 98.6 ± 0.1 | 99.2 ± 0.0 | 74.3 ± 1.1 | 94.4 ± 0.2 | 94.6 ± 0.3 | 95.0 ± 0.1 | 96.0 ± 0.5 | 89.7 ± 0.1 | 95.4 ± 0.3 | 97.5 ± 0.1 | 93.5 ± 0.3 |
| | + LAFF-C (Ours) | 96.4 ± 0.1 | 86.6 ± 0.7 | 97.1 ± 0.1 | 98.9 ± 0.1 | 95.1 ± 0.6 | 98.6 ± 0.1 | 99.2 ± 0.0 | 75.6 ± 1.1 | 94.5 ± 0.2 | 94.5 ± 0.1 | 94.2 ± 0.1 | 95.9 ± 0.5 | 88.0 ± 0.2 | 95.5 ± 0.2 | 97.5 ± 0.1 | 93.8 ± 0.3 |
| 4 | WinCLIP | 95.1 ± 0.1 | 90.8 ± 0.4 | 97.0 ± 0.1 | 98.9 ± 0.1 | 95.4 ± 0.8 | 98.8 ± 0.1 | 99.3 ± 0.0 | 81.3 ± 1.1 | 94.2 ± 0.2 | 96.6 ± 0.2 | 91.8 ± 0.1 | 98.2 ± 0.6 | 90.2 ± 0.7 | 94.8 ± 0.2 | 97.1 ± 0.1 | 94.6 ± 0.3 |
| | + LAFF-G (Ours) | 96.5 ± 0.1 | 87.6 ± 0.5 | 95.8 ± 0.2 | 98.9 ± 0.1 | 97.6 ± 0.4 | 98.6 ± 0.1 | 99.2 ± 0.0 | 76.1 ± 0.9 | 94.5 ± 0.2 | 94.7 ± 0.3 | 95.0 ± 0.0 | 97.4 ± 0.5 | 89.9 ± 0.1 | 95.5 ± 0.1 | 97.7 ± 0.1 | 94.3 ± 0.2 |
| | + LAFF-C (Ours) | 96.5 ± 0.1 | 86.8 ± 0.5 | 97.3 ± 0.1 | 98.8 ± 0.1 | 95.1 ± 0.7 | 98.6 ± 0.1 | 99.2 ± 0.0 | 77.2 ± 0.9 | 94.5 ± 0.2 | 96.5 ± 0.2 | 94.2 ± 0.1 | 97.2 ± 0.5 | 88.5 ± 0.3 | 95.5 ± 0.1 | 97.7 ± 0.1 | 94.2 ± 0.3 |
| 8 | WinCLIP | 95.2 ± 0.1 | 91.1 ± 0.3 | 97.1 ± 0.1 | 98.9 ± 0.1 | 95.5 ± 0.4 | 98.8 ± 0.1 | 99.3 ± 0.0 | 82.3 ± 0.7 | 94.3 ± 0.2 | 97.0 ± 0.4 | 91.8 ± 0.1 | 98.5 ± 0.1 | 91.2 ± 0.4 | 94.9 ± 0.2 | 97.1 ± 0.1 | 94.9 ± 0.2 |
| | + LAFF-G (Ours) | 96.6 ± 0.1 | 87.8 ± 0.4 | 95.8 ± 0.1 | 98.9 ± 0.1 | 97.8 ± 0.1 | 98.6 ± 0.1 | 99.3 ± 0.1 | 77.0 ± 0.6 | 94.4 ± 0.2 | 96.8 ± 0.3 | 95.1 ± 0.1 | 97.9 ± 0.1 | 88.8 ± 0.2 | 95.6 ± 0.1 | 97.8 ± 0.1 | 94.5 ± 0.2 |
| | + LAFF-C (Ours) | 96.6 ± 0.1 | 86.9 ± 0.4 | 97.4 ± 0.1 | 98.8 ± 0.1 | 95.3 ± 0.4 | 98.6 ± 0.1 | 99.2 ± 0.0 | 78.1 ± 0.6 | 94.6 ± 0.1 | 96.9 ± 0.3 | 94.9 ± 0.1 | 97.9 ± 0.1 | 89.0 ± 0.2 | 95.5 ± 0.1 | 97.8 ± 0.1 | 94.5 ± 0.2 |

**Table 12:** Full anomaly detection and localization AUROC (%) on the VisA dataset in few-shot settings (k-shot). We use five different sets of reference samples from the training set for each method. We **bold** the best performance and underline the second-best performance in each category.

**Anomaly Detection**

| # K | Method | Candle | Capsules | Cashew | Chewinggum | Fryum | Macaroni1 | Macaroni2 | Pcb1 | Pcb2 | Pcb3 | Pcb4 | Pipe Fryum | Average |
|---|---|---|---|---|---|---|---|---|---|---|---|---|---|---|
| 0 | WinCLIP | **96.6** | 79.1 | 92.7 | 95.9 | 74.5 | 74.6 | **65.2** | 71.1 | 43.9 | 57.5 | 84.7 | 70.6 | 75.5 |
|  | + LAFT-G (Ours) | 96.2 | 83.3 | **93.3** | 97.8 | 84.3 | **81.2** | 63.5 | 82.4 | 49.1 | 62.9 | **90.8** | 74.5 | 80.0 |
|  | + LAFT-C (Ours) | 96.1 | **84.1** | **93.3** | **98.0** | 82.5 | 80.0 | 62.5 | **84.4** | **56.2** | 67.8 | 87.6 | **74.6** | **80.6** |
| 1 | InCTRL | 84.2 ± 4.8 | 77.1 ± 2.5 | 91.1 ± 2.1 | 96.7 ± 0.4 | 93.8 ± 1.4 | 85.5 ± 3.7 | 79.0 ± 4.3 | 76.8 ± 8.8 | 74.8 ± 3.9 | 76.0 ± 4.9 | 88.3 ± 3.0 | 97.1 ± 1.6 | 85.0 ± 4.3 |
|  | WinCLIP | 97.2 ± 0.3 | 82.3 ± 1.2 | 94.1 ± 0.4 | 98.5 ± 0.2 | 90.1 ± 1.0 | 83.4 ± 1.6 | 74.5 ± 1.6 | 78.1 ± 9.7 | 61.4 ± 3.1 | 68.8 ± 3.7 | 80.8 ± 6.3 | 92.1 ± 1.6 | 83.4 ± 2.8 |
|  | + LAFT-G (Ours) | 97.5 ± 0.3 | 84.8 ± 1.3 | 94.6 ± 0.4 | 99.2 ± 0.2 | 91.8 ± 1.9 | 85.2 ± 1.4 | 72.9 ± 1.2 | 84.1 ± 0.3 | 59.0 ± 2.7 | 70.8 ± 2.6 | 89.0 ± 2.8 | 91.0 ± 1.1 | 85.0 ± 1.3 |
|  | + LAFT-C (Ours) | 97.5 ± 0.3 | 84.5 ± 1.3 | 94.6 ± 0.4 | 98.7 ± 0.2 | 91.9 ± 0.8 | 85.5 ± 1.1 | 73.0 ± 1.2 | 84.4 ± 2.8 | 64.2 ± 2.6 | 76.1 ± 2.0 | 88.0 ± 2.6 | 89.9 ± 1.3 | 85.7 ± 1.4 |
| 2 | InCTRL | 85.0 ± 4.0 | 78.9 ± 1.1 | 92.1 ± 1.6 | 97.3 ± 0.4 | 95.3 ± 0.3 | 87.9 ± 1.4 | 80.1 ± 2.9 | 81.1 ± 9.1 | 77.9 ± 2.0 | 81.7 ± 2.5 | 86.1 ± 5.7 | 97.4 ± 1.3 | 86.7 ± 2.7 |
|  | WinCLIP | 97.5 ± 0.2 | 83.9 ± 1.0 | 94.4 ± 0.5 | 98.7 ± 0.1 | 91.6 ± 0.9 | 84.9 ± 1.4 | 76.4 ± 0.9 | 83.9 ± 0.8 | 64.2 ± 2.6 | 74.8 ± 5.1 | 84.6 ± 6.0 | 92.2 ± 1.6 | 85.6 ± 1.8 |
|  | + LAFT-G (Ours) | 97.8 ± 0.2 | 86.5 ± 1.0 | 94.8 ± 0.4 | 99.2 ± 0.2 | 93.2 ± 1.0 | 86.1 ± 0.9 | 74.6 ± 0.8 | 84.9 ± 0.3 | 61.6 ± 2.1 | 72.5 ± 5.2 | 91.2 ± 3.1 | 90.9 ± 0.8 | 86.1 ± 1.3 |
|  | + LAFT-C (Ours) | 97.8 ± 0.1 | 85.9 ± 0.9 | 94.8 ± 0.5 | 99.0 ± 0.1 | 93.4 ± 1.0 | 86.3 ± 0.8 | 74.6 ± 0.9 | 86.9 ± 1.4 | 64.2 ± 1.3 | 79.7 ± 2.1 | 90.3 ± 3.1 | 90.0 ± 1.5 | 87.0 ± 1.1 |
| 4 | InCTRL | 88.5 ± 1.6 | 80.3 ± 1.4 | 92.9 ± 0.9 | 97.5 ± 0.3 | 94.9 ± 0.8 | 88.0 ± 2.7 | 81.7 ± 4.5 | 86.9 ± 3.5 | 78.2 ± 2.3 | 84.2 ± 1.8 | 89.2 ± 4.0 | 98.5 ± 0.2 | 88.4 ± 2.0 |
|  | WinCLIP | 97.7 ± 0.2 | 85.0 ± 1.1 | 95.1 ± 0.5 | 98.6 ± 0.2 | 91.9 ± 0.8 | 84.0 ± 1.6 | 77.0 ± 1.4 | 85.9 ± 3.6 | 64.3 ± 3.5 | 80.4 ± 2.2 | 88.8 ± 6.1 | 92.3 ± 1.6 | 86.8 ± 1.9 |
|  | + LAFT-G (Ours) | 98.0 ± 0.2 | 87.7 ± 1.2 | 95.5 ± 0.4 | 99.3 ± 0.1 | 92.6 ± 0.5 | 84.8 ± 0.8 | 75.2 ± 1.2 | 86.4 ± 2.8 | 61.9 ± 2.9 | 78.0 ± 2.4 | 92.8 ± 3.5 | 90.5 ± 1.5 | 87.0 ± 1.5 |
|  | + LAFT-C (Ours) | 97.9 ± 0.2 | 87.2 ± 1.1 | 95.4 ± 0.4 | 99.0 ± 0.1 | 93.4 ± 1.1 | 84.8 ± 1.1 | 75.2 ± 1.2 | 88.3 ± 2.5 | 63.5 ± 2.3 | 81.3 ± 2.3 | 92.0 ± 2.9 | 89.6 ± 1.5 | 87.4 ± 1.4 |
| 8 | InCTRL | 89.3 ± 0.8 | 81.1 ± 1.9 | 92.2 ± 1.8 | 97.7 ± 0.3 | 95.3 ± 0.7 | 89.5 ± 1.8 | 80.9 ± 4.9 | 89.8 ± 1.7 | 80.1 ± 1.6 | 85.9 ± 1.4 | 93.4 ± 2.7 | 98.1 ± 0.3 | 89.4 ± 1.7 |
|  | WinCLIP | 97.9 ± 0.0 | 85.8 ± 0.5 | 95.2 ± 0.8 | 98.7 ± 0.3 | 92.4 ± 0.8 | 84.1 ± 0.9 | 76.4 ± 2.4 | 89.4 ± 3.1 | 67.6 ± 1.2 | 82.8 ± 1.9 | 95.4 ± 1.6 | 92.6 ± 0.9 | 88.2 ± 1.2 |
|  | + LAFT-G (Ours) | 98.1 ± 0.1 | 88.5 ± 0.6 | 95.5 ± 0.6 | 99.4 ± 0.1 | 93.2 ± 1.4 | 85.0 ± 0.4 | 74.7 ± 2.3 | 90.0 ± 3.0 | 64.8 ± 0.8 | 80.5 ± 2.3 | 96.1 ± 1.3 | 91.4 ± 0.4 | 88.2 ± 1.1 |
|  | + LAFT-C (Ours) | 98.1 ± 0.1 | 87.9 ± 0.5 | 95.5 ± 0.7 | 99.0 ± 0.2 | 92.7 ± 1.0 | 84.3 ± 0.4 | 74.6 ± 2.2 | 91.3 ± 2.4 | 65.9 ± 0.6 | 83.1 ± 1.9 | 96.2 ± 1.4 | 89.9 ± 0.8 | 88.3 ± 1.0 |

**Anomaly Localization**

| # K | Method | Candle | Capsules | Cashew | Chewinggum | Fryum | Macaroni1 | Macaroni2 | Pcb1 | Pcb2 | Pcb3 | Pcb4 | Pipe Fryum | Average |
|---|---|---|---|---|---|---|---|---|---|---|---|---|---|---|
| 0 | WinCLIP | 87.1 | 70.0 | 81.9 | 95.1 | 82.1 | 57.4 | 53.3 | 41.5 | 58.8 | 69.4 | 91.4 | 80.5 | 72.4 |
|  | + LAFT-G (Ours) | 90.4 | 72.7 | 81.3 | 97.0 | 87.6 | 84.4 | 77.6 | 79.0 | 82.6 | 73.2 | 90.2 | 87.4 | 83.6 |
|  | + LAFT-C (Ours) | 90.0 | 70.3 | 81.5 | 95.4 | 89.7 | 88.6 | 79.9 | 74.5 | 78.8 | 70.8 | 91.7 | 85.9 | 83.1 |
| 1 | WinCLIP | 96.5 ± 0.3 | 93.2 ± 0.1 | 96.5 ± 0.5 | 99.1 ± 0.0 | 91.4 ± 0.6 | 93.2 ± 0.8 | 92.9 ± 0.8 | 96.2 ± 0.4 | 91.5 ± 0.6 | 92.1 ± 0.8 | 94.4 ± 0.5 | 95.9 ± 0.5 | 94.4 ± 0.5 |
|  | + LAFT-G (Ours) | 96.0 ± 0.3 | 88.0 ± 0.4 | 95.2 ± 0.5 | 98.6 ± 0.0 | 92.4 ± 0.5 | 91.4 ± 0.5 | 91.3 ± 0.8 | 94.9 ± 0.4 | 89.4 ± 0.7 | 86.6 ± 1.1 | 95.7 ± 0.3 | 95.4 ± 0.4 | 92.9 ± 0.5 |
|  | + LAFT-C (Ours) | 96.0 ± 0.3 | 87.7 ± 0.3 | 95.2 ± 0.5 | 99.1 ± 0.0 | 93.0 ± 0.9 | 89.9 ± 0.6 | 91.2 ± 0.8 | 96.3 ± 0.3 | 92.1 ± 0.4 | 84.8 ± 0.8 | 94.9 ± 0.2 | 95.9 ± 0.4 | 93.0 ± 0.5 |
| 2 | WinCLIP | 96.5 ± 0.2 | 93.4 ± 0.2 | 96.6 ± 0.2 | 99.1 ± 0.0 | 91.8 ± 0.4 | 93.7 ± 0.5 | 93.1 ± 0.6 | 96.8 ± 0.5 | 92.1 ± 0.3 | 93.4 ± 0.6 | 95.4 ± 0.4 | 95.8 ± 0.3 | 94.8 ± 0.4 |
|  | + LAFT-G (Ours) | 95.9 ± 0.2 | 88.3 ± 0.2 | 95.3 ± 0.2 | 98.6 ± 0.0 | 92.5 ± 0.2 | 91.8 ± 0.4 | 91.4 ± 0.6 | 95.4 ± 0.2 | 90.0 ± 0.4 | 93.1 ± 0.6 | 96.1 ± 0.2 | 95.4 ± 0.3 | 93.7 ± 0.3 |
|  | + LAFT-C (Ours) | 96.0 ± 0.2 | 87.9 ± 0.3 | 95.2 ± 0.2 | 99.1 ± 0.0 | 94.0 ± 0.5 | 90.3 ± 0.4 | 91.2 ± 0.6 | 97.0 ± 0.5 | 92.6 ± 0.3 | 85.5 ± 0.7 | 95.2 ± 0.1 | 95.8 ± 0.3 | 93.3 ± 0.3 |
| 4 | WinCLIP | 96.6 ± 0.1 | 93.7 ± 0.1 | 96.7 ± 0.1 | 99.1 ± 0.0 | 92.1 ± 0.1 | 93.1 ± 0.5 | 92.9 ± 0.8 | 97.3 ± 0.7 | 92.7 ± 0.2 | 94.1 ± 0.2 | 96.0 ± 0.4 | 95.8 ± 0.3 | 95.0 ± 0.3 |
|  | + LAFT-G (Ours) | 96.0 ± 0.1 | 88.3 ± 0.3 | 95.2 ± 0.1 | 98.6 ± 0.0 | 92.5 ± 0.1 | 91.0 ± 0.3 | 91.1 ± 0.8 | 96.3 ± 0.7 | 90.4 ± 0.3 | 93.8 ± 0.3 | 96.3 ± 0.2 | 95.5 ± 0.3 | 93.8 ± 0.3 |
|  | + LAFT-C (Ours) | 96.2 ± 0.1 | 87.9 ± 0.3 | 95.3 ± 0.1 | 98.7 ± 0.0 | 94.3 ± 0.3 | 89.6 ± 0.4 | 91.0 ± 0.6 | 97.5 ± 0.7 | 93.1 ± 0.2 | 92.0 ± 0.3 | 95.3 ± 0.1 | 95.8 ± 0.3 | 93.9 ± 0.3 |
| 8 | WinCLIP | 96.7 ± 0.1 | 93.9 ± 0.3 | 96.8 ± 0.1 | 99.1 ± 0.0 | 92.4 ± 0.2 | 93.0 ± 0.4 | 92.4 ± 1.6 | 97.9 ± 0.8 | 93.3 ± 0.3 | 94.7 ± 0.3 | 96.7 ± 0.2 | 95.9 ± 0.2 | 95.2 ± 0.4 |
|  | + LAFT-G (Ours) | 96.2 ± 0.1 | 88.4 ± 0.3 | 95.4 ± 0.1 | 98.3 ± 0.0 | 92.5 ± 0.1 | 90.5 ± 0.2 | 90.5 ± 1.7 | 97.2 ± 0.8 | 91.0 ± 0.3 | 94.4 ± 0.3 | 96.8 ± 0.2 | 95.7 ± 0.2 | 93.9 ± 0.4 |
|  | + LAFT-C (Ours) | 96.2 ± 0.1 | 87.9 ± 0.3 | 95.3 ± 0.1 | 99.1 ± 0.0 | 94.8 ± 0.2 | 92.9 ± 0.1 | 90.4 ± 1.7 | 98.0 ± 0.6 | 93.5 ± 0.1 | 92.6 ± 0.4 | 96.8 ± 0.2 | 95.9 ± 0.2 | 94.5 ± 0.3 |

# E   ALGORITHM

The following pseudocode demonstrates the implementation of LAFT AD, using a syntax similar to NumPy, as the notation used in (Radford et al., 2021).

```
# model: the CLIP model
# prompts: the list of prompts provided by the user
# train_images: the collection of normal images
# test_images: the collection of images to be tested
# d: the number of principle axis

# Compute attribute subspace
text_features = model.encode_text(prompts)
pair_diffs = pairwise_difference(text_features)
basis = pca(pair_diffs, d)

# Encode images
train_features = model.encode_image(train_images)
test_features  = model.encode_image(test_images)

# Guide
train_laft_features = inner_projection(train_features, basis)
test_laft_features  = inner_projection(test_features,  basis)

anomaly_scores = knn(train_laft_features, test_laft_features)

# Ignore
train_laft_features = orthogonal_projection(train_features, basis)
test_laft_features  = orthogonal_projection(test_features,  basis)

anomaly_scores = knn(train_laft_features, test_laft_features)
```

