# OpenReview forum: "Language-Assisted Feature Transformation for Anomaly Detection"
_ICLR.cc/2025/Conference — ICLR 2025 Poster_

### Official Review · Reviewer_dr7M · 2024-10-20

**Soundness:** 3
**Presentation:** 3
**Contribution:** 3
**Rating:** 6
**Confidence:** 5

**Summary:**

On the basis of existing anomaly detection methods based on visual language alignment, this paper proposes using task related languages for task oriented feature information screening and transformation to improve the model's anomaly detection capability. The experiment was conducted on multiple datasets and demonstrated better performance compared with existing methods.

**Strengths:**

1. This paper is with clear motivation.
2. This paper is well-organized and easy to follow.

**Weaknesses:**

1. The criteria for selecting text prompts are ambiguous. Some datasets utilize the category names of the samples, while others employ diverse descriptions. These approaches rest on the critical assumption that anomalies are distinctly defined, exemplified by MNIST, where anomalies arise from differences in numerals rather than variations in handwriting styles or colors. Should the actual anomalies diverge from these presuppositions, might the proposed model's performance diminish relative to methods devoid of textual guidance? In other words, could the model forfeit its capacity to detect all possible anomalies?

2. In the MVTec dataset experiment, the author opted not to employ the concise anomaly descriptions provided by the dataset itself for text prompts, instead relying solely on item categories, mirroring the approach of WinCLIP. What rationale informed this decision?

3. The proposed model is an extension of WinCLIP, yet it appears to forgo the anomaly segmentation functionality inherent to WinCLIP. Is this omission attributable to certain design elements that potentially diminish the model's anomaly localization capabilities?

4. Experiments have been conducted on synthetic datasets like MNIST and CelebA by altering the original datasets. While I acknowledge the challenge of selecting appropriate text prompts for real-world datasets such as MVTec, the author should endeavor to incorporate more authentic datasets into their study, such as the VisA dataset utilized in WinCLIP or the medical AD benchmark employed in MVFA [a].

[a] Adapting Visual-Language Models for Generalizable Anomaly Detection in Medical Images. CVPR 2024.

**Questions:**

See the weakness.

---

### Official Review · Reviewer_23Gv · 2024-10-28

**Soundness:** 3
**Presentation:** 2
**Contribution:** 3
**Rating:** 8
**Confidence:** 3

**Summary:**

This paper proposes a feature transformation methodology using concept axes, which are the principal components of the difference vectors between text embeddings of prompts specially designed to ignore nuisance attributes/highlight important attributes for anomaly detection.

**Strengths:**

The methodology is interesting and a solid contribution to this direction of research in vision-language modelling for anomaly detection.

The results appear to be promising in the experiments presented, although a wider range of experimental setups would be more convincing (see weakness)

The ablation study is comprehensive.

**Weaknesses:**

1. Figure 1 is not particularly intuitive or clear, and it is not explained in the text.

2. As the exact formulation of prompts is absolutely critical for this methodology, it should have more dedicated explanation in the main text of the paper, not relegated almost entirely to the appendix.

3. There are not many baselines, and it would have been more convincing if you compare more baselines with and without LAFT transformations.

4. The range of experiments presented are quite restricted. For example with Coloured MNIST, it appears that only one number-colour combination as the normal set was tried. It would be more proper to conduct multiple experiments with different combinations of attributes and show the average result. The same can be said for the other datasets.

**Questions:**

Please address the points raised in the Weakness section. Also:

1. What is the purpose of including Aux. prompts?

2. How does different CLIP architecture and also different VLMs affect performance?

---

> ### Comment · Reviewer_DtcB · 2024-11-24
>
> Thanks the authors for the detailed response. I've checked the authors response as well as the other reviewers' comment. I am still leaning towards a rejection as the newly added experiments clearly show an unsatisfied performance. I will keep my original score.

---

> > ### Author Response · Authors · 2024-11-25
> >
> > Thank you for taking the time to reply. Since this is reviewer 23Gv's thread, we'll continue to answer in your thread.

---

### Official Review · Reviewer_DtcB · 2024-11-02

**Soundness:** 3
**Presentation:** 3
**Contribution:** 3
**Rating:** 5
**Confidence:** 5

**Summary:**

The paper introduces Language-Assisted Feature Transformation (LAFT), a novel framework that leverages vision-language models (like CLIP) to enhance anomaly detection. Traditional anomaly detection methods often struggle to capture user-defined nuances of normality, particularly when attributes are entangled or datasets are incomplete. LAFT tackles this by enabling feature transformations guided by natural language prompts. These prompts align visual features with user intent by projecting image features onto specific concept subspaces within a shared embedding space. The paper also proposes LAFT AD, a k-nearest-neighbor (kNN)-based method combining LAFT with anomaly detection, and extends this work into WinCLIP+LAFT, designed for industrial applications. The effectiveness of LAFT is demonstrated across datasets like Colored MNIST, Waterbirds, CelebA, and MVTec AD, showing superior performance in both semantic and industrial anomaly detection.

**Strengths:**

1. LAFT bridges a gap in anomaly detection by allowing users to express preferences using natural language, providing more control over what is considered "normal."
2. Unlike other feature transformation models, LAFT does not require additional training, making it efficient for settings with scarce data.
3. The experimental results demonstrate that LAFT outperforms state-of-the-art methods, particularly in semantic anomaly detection tasks.

**Weaknesses:**

1. While LAFT demonstrates significant improvements in controlled environments, such as the Colored MNIST dataset, its performance gains appear less pronounced when applied to complex real-world datasets. This discrepancy suggests that the model may struggle to maintain robustness across multiple intricate attributes, highlighting the need for further refinement in handling multi-attribute scenarios.
2. The experimental setup lacks comprehensive comparisons, particularly between language-assisted and vision-assisted approaches. For instance, incorporating image guidance by utilizing related reference normal images (e.g., normal digits in various colors) or color-augmentation for kNN baseline could provide valuable insights. A thorough examination of both language-based and vision-based assistance would strengthen the evaluation of LAFT's efficacy.
3. The impact of the number of PCA components, which is the sole hyperparameter in LAFT, is not adequately investigated. Given that this parameter influences the model's performance, it is crucial to explore its effect across different datasets. Specifically, an analysis of whether a larger number of components may be beneficial for more complex datasets would provide valuable insights into optimizing the model’s performance.

**Questions:**

1. In Table 8, the header refers to "bird," which is inconsistent with the title of the Colored MNIST dataset mentioned (maybe a typo). Could the authors clarify this discrepancy?
2. What are the sizes of the training sets for each dataset used in the experiments? Given that these samples serve as candidates for kNN search, how might the number of training samples affect the final performance of the model?
3. The experimental results on the MVTec AD dataset in Table 3 suggest that InCTRL might outperform WinCLIP+LAFT when considering deviation, especially when the number of shots exceeds 2. Could the authors provide detailed experimental results for each of the five different reference sample sets?

---

### Official Review · Reviewer_P8a7 · 2024-11-04

**Soundness:** 3
**Presentation:** 2
**Contribution:** 2
**Rating:** 5
**Confidence:** 5

**Summary:**

The paper introduces a feature transformation method aimed at focusing on specific image attributes guided by language. The approach, termed Language-Assisted Feature Transformation (LAFT), leverages the shared embedding space of vision-language models (specifically CLIP) to modify image features according to user-defined concepts expressed in natural language, enabling enhanced anomaly detection capabilities without additional training.

**Strengths:**

- The authors explore a valuable research topic that contributes to the current body of knowledge—how to adjust decision boundaries using language to enhance CLIP’s anomaly detection performance.
- The proposed method stands out due to its training-free nature, which provides flexibility in application across various tasks with limited data.

**Weaknesses:**

- The paper uses the vector difference between two textual descriptions to represent a single attribute and maps this attribute directly to image feature transformation. However, this simplification raises at least three issues:
   - The properties of objects cannot be adequately represented by the difference between two concepts.
   - Real-world attributes are often complex and may involve different colors or textures across various parts of an object.
   - The text embedding space and the image embedding space in CLIP are not perfectly aligned; therefore, vectors derived from the text space may not be directly applicable to the image space.

- To validate the effectiveness of feature transformation, using a CLIP-based classification task would be more suitable than anomaly detection.

- The paper lacks results on anomaly localization, which is crucial for industrial applications.

- The language throughout the paper could be clearer. It is recommended to refer to previous works using proper method names and provide concise descriptions of these methods.

- The axis labels in Figure 3 are inconsistent. How were the attributes 'Number' and 'Color' derived?

- The dataset chosen for experiments, SEMANTIC ANOMALY DETECTION, focuses on distinguishing simple concepts. Why not test the method on widely recognized OOD datasets such as ImageNet-1k and OpenOOD? Industrial anomaly detection would benefit from validation on datasets like VisA and Real-IAD as well.

- The comparison methods included are relatively weak. Why not compare with more recent OOD detection approaches such as NegLabel [1] and ClipN [2]?
---
- \[1] X. Jiang, F. Liu, Z. Fang, H. Chen, T. Liu, F. Zheng, and B. Han, “Negative label guided OOD detection with pretrained vision-language models,” in The Twelfth International Conference on Learning Representations, 2024.
- \[2] Hualiang Wang, Yi Li, Huifeng Yao, and Xiaomeng Li. ClipN for zero-shot OOD detection: Teaching CLIP to say no. ICCV, 2023.
---
If the author can address my concerns, I will consider increasing the score.

**Questions:**

1. What does $c_i$ represent in Equations 5 and 6?
2. For zero-shot anomaly detection, can the transformed image features still match the text features effectively?

---

### Meta-Review · Area_Chair_h4pN · 2024-12-15

**Metareview:**

This paper proposes LAFT, a method for anomaly detection that uses natural language guidance to transform features in a vision-language embedding space. The approach is training-free and shows strong performance on semantic anomaly detection tasks while enabling user-defined customization of detection boundaries. Strengths include the innovative use of language guidance, robustness demonstrated through comprehensive experiments, and practical applicability in scenarios with limited data. However, concerns about scalability to complex real-world datasets and limited comparisons with alternative baselines were noted. The authors addressed these issues during the rebuttal phase, adding new baselines and analyses. This paper is recommended for Accept (poster) due to its novel contributions and practical utility.

**Additional Comments On Reviewer Discussion:**

Reviewers expressed concerns about robustness on industrial datasets, limited evaluation of multi-attribute scenarios, and theoretical justification of the feature transformation process. The authors addressed these by adding experiments with industrial datasets like VisA, expanding comparisons with baselines such as CLIPN, and providing detailed explanations of LAFT’s mechanism. While some concerns persist regarding scalability and applicability to broader contexts, reviewers acknowledged the significant improvements during the discussion phase, resulting in adjusted scores.

---

### Decision · Program_Chairs · 2025-01-22

Accept (Poster)